# BRCA1/BRC-1 and SMC-5/6 regulate DNA repair pathway engagement during *Caenorhabditis elegans* meiosis

Erik Toraason[1], Alina Salagean[1], David E Almanzar[2], Jordan E Brown[1], Colette M Richter[1], Nicole A Kurhanewicz[1], Ofer Rog[2], Diana E Libuda[1]*

[1]Institute of Molecular Biology, Department of Biology, University of Oregon, Eugene, United States; [2]School of Biological Sciences and Center for Cell and Genome Sciences, University of Utah, Salt Lake City, United States

**Abstract** The preservation of genome integrity during sperm and egg development is vital for reproductive success. During meiosis, the tumor suppressor BRCA1/BRC-1 and structural maintenance of chromosomes 5/6 (SMC-5/6) complex genetically interact to promote high fidelity DNA double strand break (DSB) repair, but the specific DSB repair outcomes these proteins regulate remain unknown. Using genetic and cytological methods to monitor resolution of DSBs with different repair partners in *Caenorhabditis elegans*, we demonstrate that both BRC-1 and SMC-5 repress intersister crossover recombination events. Sequencing analysis of conversion tracts from homolog-independent DSB repair events further indicates that BRC-1 regulates intersister/intrachromatid noncrossover conversion tract length. Moreover, we find that BRC-1 specifically inhibits error prone repair of DSBs induced at mid-pachytene. Finally, we reveal functional interactions of BRC-1 and SMC-5/6 in regulating repair pathway engagement: BRC-1 is required for localization of recombinase proteins to DSBs in *smc-5* mutants and enhances DSB repair defects in *smc-5* mutants by repressing theta-mediated end joining (TMEJ). These results are consistent with a model in which some functions of BRC-1 act upstream of SMC-5/6 to promote recombination and inhibit error-prone DSB repair, while SMC-5/6 acts downstream of BRC-1 to regulate the formation or resolution of recombination intermediates. Taken together, our study illuminates the coordinated interplay of BRC-1 and SMC-5/6 to regulate DSB repair outcomes in the germline.

*For correspondence: dlibuda@uoregon.edu

Competing interest: The authors declare that no competing interests exist.

## Editor's evaluation

This study provides an important insight in the regulation of DSB repair during meiosis in *C. elegans*. The analysis is based on genetic, genomic and cytological approaches allowing to monitoring distinct DNA repair pathways and providing compelling evidence for their regulation and genetic controls.

## Introduction

Meiosis is the specialized form of cell division by which most sexually reproducing organisms generate haploid gametes. In a diploid organism, each meiotic cell begins prophase I with four copies of the genome – two homologous chromosomes (homologs) and identical copies of each homolog called sister chromatids. As mutations incurred in the gamete genome will be passed on to the resultant progeny, it is crucial that genome integrity be maintained during meiosis. Despite this risk, a highly conserved feature of the meiotic program is induction of DNA double strand breaks (DSBs) by the topoisomerase-like protein Spo11 (*Keeney et al., 1997*; *Bergerat et al., 1997*). A limited subset of

DSBs must engage the homologous chromosome as a recombination partner and be repaired as a crossover event to forge a physical connection between homologs that facilitates accurate chromosome segregation at the meiosis I division. DSBs are incurred in excess of the number of eventual crossovers and other pathways must therefore be utilized to repair residual DSBs. How meiotic cells regulate repair pathway engagement to both accurately and efficiently resolve DSBs is a critical question in the field of genome integrity.

In many organisms, the majority of meiotic DSBs are repaired through interhomolog noncrossover recombination mechanisms (*Hunter, 2015*). Multiple models are proposed for how meiotic noncrossover repair occurs. Evidence in *Drosophila* suggests that both interhomolog noncrossovers and crossovers may be generated by differential processing of similar joint molecule intermediates (*Crown et al., 2014*). Work in budding yeast, mammals, and *Arabidopsis* indicates that the majority of interhomolog noncrossovers are generated via synthesis-dependent strand annealing (SDSA) with the homolog (*Hunter, 2015*). In SDSA, one or both resected end(s) of the DSB invades a repair template, new sequence is synthesized, the strand dissociates from its repair template, and finally utilizes the synthesized sequence to anneal to the other resected end of the DSB.

Meiotic DSBs may also be repaired by recombination with the sister chromatid (*Goldfarb and Lichten, 2010*; *Toraason et al., 2021c*; *Almanzar et al., 2021*; *Schwacha and Kleckner, 1997*). In budding yeast, DSB repair by intersister recombination is disfavored so as to promote recombination with the homologous chromosome (*Goldfarb and Lichten, 2010*; *Schwacha and Kleckner, 1997*; *Humphryes and Hochwagen, 2014*; *Kim et al., 2010*; *Schwacha and Kleckner, 1994*). In metazoan meiosis, however, the engagement of intersister repair has proven challenging to detect and quantify. While recombination between polymorphic homologs may be readily studied via sequence conversions in final repair products, the identical sequences of sister chromatids preclude the detection of intersister recombination by sequencing-based approaches. Recently, two methods have been developed in the nematode *Caenorhabditis elegans* to enable direct detection of homolog-independent meiotic recombination (*Toraason et al., 2021c*; *Almanzar et al., 2021*). *Toraason et al., 2021c* constructed an intersister/intrachromatid repair (ICR) assay that exploits nonallelic recombination at a known locus in the genome to identify homolog-independent repair events in resultant progeny (*Figure 1-figure supplement 1 A*). *Almanzar et al., 2021* designed an EdU labeling assay to cytologically identify sister chromatid exchanges (SCEs) in compacted chromosomes at diakinesis. Together, these studies demonstrated that: (1) homolog-independent meiotic recombination occurs in *C. elegans*; (2) the sister chromatid and/or same DNA molecule is the exclusive recombination repair template in late prophase I; and, (3) intersister crossovers are rare and represent a minority of homolog-independent recombination products (*Toraason et al., 2021c*; *Almanzar et al., 2021*).

While meiotic cells primarily utilize homologous recombination to resolve DSBs, error prone repair pathways are also available in meiosis to repair DSBs at the risk of introducing de novo mutations (*Gartner and Engebrecht, 2022*). These error prone mechanisms are repressed in wild type contexts to promote recombination repair but are activated in mutants that disrupt recombination (*Macaisne et al., 2018*; *Yin and Smolikove, 2013*; *Kamp et al., 2020*; *Lemmens et al., 2013*). Non-homologous end joining (NHEJ), which facilitates the ligation of blunt DNA ends by the DNA ligase IV homolog LIG-4, is active in the *C. elegans* germline (*Macaisne et al., 2018*; *Yin and Smolikove, 2013*). Recent studies have revealed that microhomology-mediated end-joining facilitated by the DNA polymerase θ homolog POLQ-1 (theta-mediated end-joining, TMEJ) is the primary pathway by which small mutations are incurred in *C. elegans* germ cells and somatic cells (*Kamp et al., 2020*; *van Schendel et al., 2015*). Since neither NHEJ nor TMEJ are required for successful meiosis (*Kamp et al., 2020*; *Lemmens et al., 2013*; *Volkova et al., 2020*; *Colaiácovo et al., 2003*), recombination is likely sufficient for meiotic DSB repair and gamete viability under normal conditions.

The structural maintenance of chromosomes 5/6 complex and tumor suppressor BRCA1 (SMC-5/6 and BRC-1 respectively in *C. elegans*) are highly conserved and regulate meiotic DSB repair in *C. elegans* (*Kamp et al., 2020*; *Bickel et al., 2010*; *Hong et al., 2016*; *Li et al., 2018*; *Janisiw et al., 2018*; *Odiba et al., 2024*). The SMC-5/6 complex is vital for preservation of meiotic genome integrity, as *C. elegans* mutants for *smc-5* exhibit a transgenerational sterility phenotype (*Bickel et al., 2010*). Although null mutations of *smc-5, smc-6,* and *brc-1* revealed that they are not required for development nor reproduction in *C. elegans* (*Bickel et al., 2010*; *Li et al., 2018*; *Janisiw et al., 2018*; *Adamo et al., 2008*), both SMC-5/6 and BRC-1 are required for efficient DSB repair, as *smc-5* and *brc-1* null

mutants both display meiotic chromosome fragmentation at diakinesis indicative of unrepaired DSBs (***Bickel et al., 2010***). In *C. elegans,* BRC-1 has also been shown to repress error prone DSB repair via NHEJ and TMEJ (***Kamp et al., 2020***; ***Li et al., 2020***), and acts in parallel with the tumor suppressor 53BP1/HSR-9 (***Hariri et al., 2023***). Further, SMC-5/6 and BRC-1 may promote genome integrity in part by facilitating efficient recombination, as *smc-5* and *brc-1* mutants exhibit persistent DSBs marked by the recombinase RAD-51 (***Kamp et al., 2020***; ***Bickel et al., 2010***; ***Adamo et al., 2008***; ***Boulton et al., 2004***), suggesting that early recombination steps are disrupted in these mutants. BRC-1 further prevents recombination between heterologous templates to promote accurate recombination repair (***León-Ortiz et al., 2018***). Despite these apparent DNA repair defects, interhomolog crossover formation is largely unaffected by *smc-5* and *brc-1* mutations (***Bickel et al., 2010***; ***Li et al., 2018***; ***Janisiw et al., 2018***; ***Adamo et al., 2008***). Taken together, these data have contributed to the model that that SMC-5/6 and BRC-1 are required for intersister repair in *C. elegans.*

SMC-5/6 and BRC-1 genetically interact to regulate germline DSB repair. The incidence of unrepaired DSBs observed in both *smc-5* and *brc-1* single mutants are not additive in the double *smc-5;brc-1* mutant context, which suggests that SMC-5/6 and BRC-1 may share some DSB repair functions (***Bickel et al., 2010***). Other experiments, however, indicate opposing functions for SMC-5/6 and BRC-1, as both the mitotic DNA replication defects in *smc-5* mutants and the synthetic lethality of *smc-5;him-6* (BLM helicase) double mutants are suppressed by *brc-1* mutation (***Hong et al., 2016***; ***Wolters et al., 2014***). Crucially, the specific steps of recombination regulated by SMC-5/6 and BRC-1 that intersect to influence DNA repair outcomes remain unknown.

To determine the DSB repair functions of SMC-5 and BRC-1 that regulate DNA repair outcomes during *C. elegans* meiosis, we employed a multipronged approach utilizing genetic assays, cytology, sequence analysis of recombinant loci, and functional DSB repair assays in *smc-5* and *brc-1* mutants. We find that SMC-5 and BRC-1 function to repress meiotic intersister crossover recombination, and that BRC-1 specifically regulates homolog-independent noncrossover intermediate processing. Through these experiments, we also find that BRC-1 prevents mutagenic DSB repair at the mid-pachytene stage of meiotic prophase I. By assessing germ cell capacity to resolve exogenous DSBs, we demonstrate that meiotic nuclei become more dependent on BRC-1 for DSB repair in late stages of meiotic prophase I. Finally, we reveal that *smc-5* mutant DSB repair defects are suppressed by loss of *brc-1*, which impedes gamete viability in part by repressing error prone repair pathways and promoting recombination. Taken together, our study defines specific functions and interactions of BRC-1 and SMC-5 that regulate meiotic DSB repair outcomes across meiotic prophase I.

## Results
### BRC-1 restricts intersister crossovers

To directly assess the functions of BRC-1 in homolog-independent DSB repair, we employed the recently developed intersister/intrachromatid (ICR) assay (***Toraason et al., 2021c***; ***Toraason et al., 2021b***). The ICR assay enables: (1) the controlled generation of a single DSB in *C. elegans* meiotic nuclei via heat shock inducible mobilization of a Mos1 transposon (***Bessereau et al., 2001***; ***Robert and Bessereau, 2007***); (2) detection of the repair outcome of the induced DSB with the sister chromatid or same DNA molecule by reconstituting GFP fluorescence in resultant progeny; and, (3) delineation of homolog-independent crossover and noncrossover recombination outcomes (***Toraason et al., 2021c***; ***Figure 1—figure supplement 1A***). Since the *C. elegans* germline is organized in a spatial-temporal gradient in which nuclei move progressively through the stages of meiotic prophase I along the distal-proximal axis (***Rosu et al., 2011***; ***Jaramillo-Lambert et al., 2007***; ***Cahoon and Libuda, 2021***), oocytes at all stages of meiotic prophase I can be affected simultaneously by a specific treatment, such as heat shock or irradiation. As the rate of meiotic progression in the *C. elegans* germline is known (***Rosu et al., 2011***; ***Jaramillo-Lambert et al., 2007***; ***Cahoon and Libuda, 2021***), we can score resultant progeny at specific timepoints post heat shock to distinguish oocytes that incurred a Mos1-excision induced DSB at the stages of prophase I when the homologous chromosome is available as a repair partner (the 'interhomolog window', leptotene-mid pachytene, 22–58 hr post heat shock) from the stages when the homolog is not readily engaged for DSB repair (the 'non-interhomolog window', late pachytene-diplotene, 10–22 hr post heat shock) (***Rosu et al., 2011***).

We performed the ICR assay in a *brc-1(xoe4)* mutant, which removes the entire *brc-1* coding sequence (*Li et al., 2018*). If BRC-1 is required for efficient intersister repair, then we expected the overall frequency of ICR assay GFP+ recombinant progeny to be reduced. Contrary to this hypothesis, we found that GFP+ progeny were elevated at all interhomolog window timepoints and were not reduced within the non-interhomolog window (*Figure 1—figure supplement 2A*). Since the ICR assay restricts DSB recombination repair to homolog-independent repair and should therefore represent the 'ceiling' for recombination frequencies, this increase in GFP+ recombinant progeny was unexpected. Given that BRC-1 represses heterologous recombination (*León-Ortiz et al., 2018*) and that polymorphisms, such as those contained in the GFP repair template in the ICR assay (*Toraason et al., 2021c*), reduce the likelihood of recombination between templates, we hypothesized that this increase in GFP+ recombinants may represent relaxed bias for the allelic repair template in the ICR assay. However, *msh-2* mutants, which are defective in heteroduplex recognition and rejection (*Chakraborty and Alani, 2016*), did not display increased ICR assay recombinants (*Figure 1—figure supplement 2C*), suggesting that altered repair template bias does not underpin this effect. Increased recombinants could also be derived from elevated rates of Mos1 excision; however, we did not observe an overall increase in recombinant progeny in a parallel Mos1-based assay in *brc-1* mutants (*Figure 1—figure supplements 1B and 3A*), which suggests that Mos1 does not excise more frequently in *brc-1* mutants. As we cannot be certain of the source of increased recombinants in the *brc-1* ICR assay, we refrain from making any inference that requires comparing the absolute levels of homolog-independent recombination events in our *brc-1* ICR assay data. Regardless of the absolute number of ICR assay GFP+ recombinant progeny, we identified both crossover and noncrossover recombinants in the interhomolog window (*Figure 1—figure supplement 2A*), demonstrating that BRC-1 alone is not essential for intersister/intrachromatid crossover or noncrossover repair. Notably, the overall proportion of crossover progeny among recombinants identified was increased at all timepoints scored for the *brc-1(xoe4)* null mutant (*Figure 1A*), which is similar to the increase in the proportion of ICR assay crossover events we also found in the classical *brc-1(tm1145);brd-1(dw1)* mutants (*Figure 1—figure supplement 2B*). Taken together, these data collectively suggest that BRC-1 functions in *C. elegans* meiosis to repress intersister/intrachromatid crossover events.

To confirm that intersister crossovers are indeed more frequent in a *brc-1* mutant, we employed a recently developed cytological assay that utilizes EdU incorporation to visualize sister chromatid exchanges (SCEs) in compacted diakinesis chromosomes (*Figure 1B*; *Almanzar et al., 2021*; *Almanzar et al., 2022*). Notably, this cytological assay detects SCEs from endogenous SPO-11 induced DSBs. While SCEs are found in only 4.1% of chromatid pairs in a wild type background (2/49 chromatid pairs scored, 95% Binomial CI 1.1–13.7%; *Almanzar et al., 2021*), we detected SCEs at an elevated rate of 19.2% in a *brc-1(xoe4)* mutant (*Figure 1B–C*, 5/26 chromatid pairs scored, 95% Binomial CI 8.5–37.9%, Fisher's Exact Test p=0.045). When we compared the levels of SCEs cytologically identified with the frequency of ICR assay crossovers generated from Mos1-induced DSBs within the interhomolog window, the elevated frequency of SCEs (4.7-fold increase) closely mirrored the relative increase in crossovers as a proportion of all recombinants observed in the *brc-1* mutant ICR assay (4.6-fold increase). Taken together, these results demonstrate that BRC-1 functions to repress intersister crossover recombination during *C. elegans* meiosis for both SPO-11- and Mos1-induced DSBs.

## BRC-1 is not required for interhomolog recombination

Since BRC-1 acts to repress crossover recombination between sister chromatids, we next assessed if *brc-1* mutants exhibit defects in interhomolog recombination, including interhomolog crossovers. To assess the overall rates of interhomolog noncrossover and crossover recombination, we employed an established interhomolog (IH) recombination assay (*Figure 1—figure supplement 1B*; *Rosu et al., 2011*) that enables: (1) controlled generation of a single DSB in meiotic nuclei via heat-shock inducible Mos1 excision (*Robert and Bessereau, 2007*; *Rosu et al., 2011*); (2) identification of interhomolog DSB repair of the induced DSB by reversion of an uncoordinated movement 'Unc' phenotype (non-Unc progeny, see Materials and methods); and (3) delineation of interhomolog noncrossover and crossover repair outcomes (see Materials and methods). Notably, DSB repair in the IH assay that produces in-frame insertions or deletions can also yield non-Unc progeny that are phenotypically indistinguishable from noncrossover recombinants (*Robert et al., 2008*). While mutagenic repair in the IH assay is rare in a wild type context (*Robert et al., 2008*), *brc-1* mutants incur small mutations

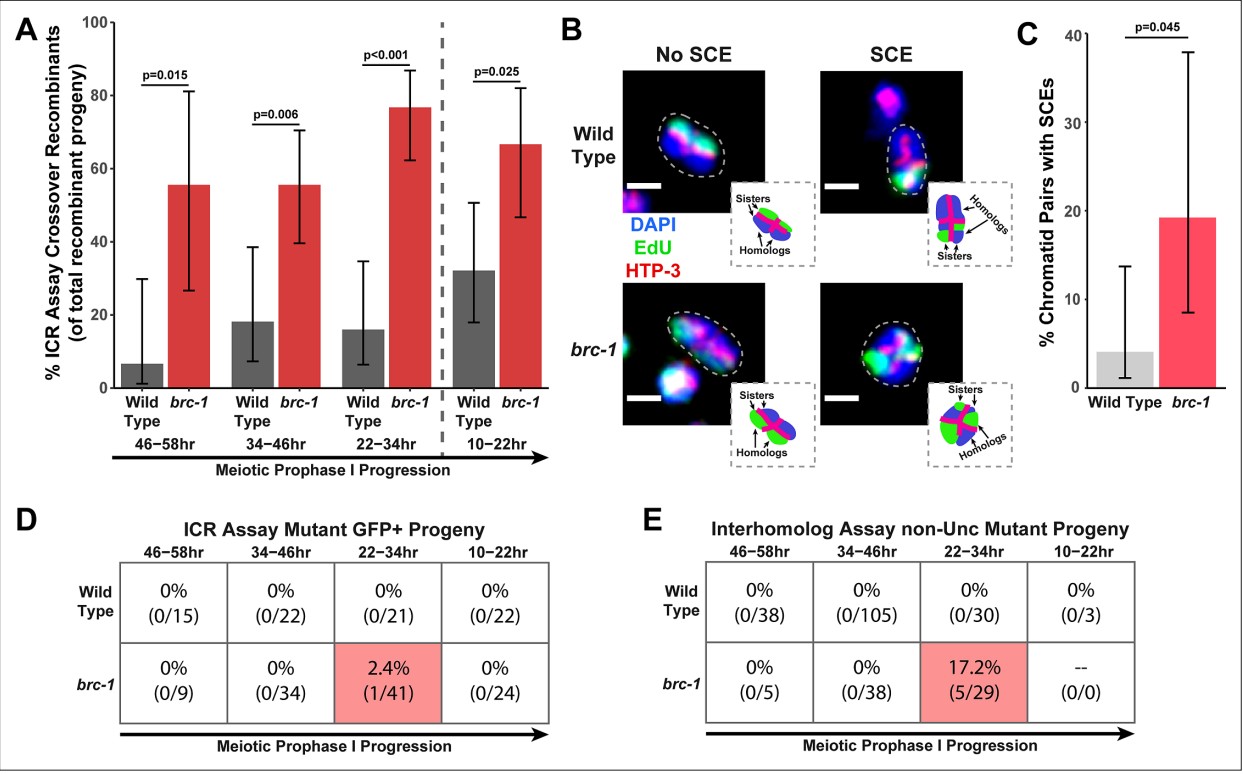

**Figure 1.** BRC-1 represses intersister crossovers and error-prone repair. (**A**) Bar plot displaying the percent of crossover recombinant progeny identified in wild type and *brc-1* ICR assays out of all recombinant progeny scored. Frequencies of recombinants identified overall in ICR assays is displayed in *Figure 1—figure supplement 2A*. Wild type ICR assay data is shared in **A** and *Figures 2A, B and 4A*, *Figure 1—figure supplement 2*, *Figure 2—figure supplement 1*, and *Figure 4—figure supplement 1*. *brc-1* ICR assay data is shared in **A** and *Figure 4A*, *Figure 1—figure supplement 2*, and *Figure 4—figure supplement 1*. The total number of recombinant progeny scored in each dataset are (10–22 hr/22–34 hr/34–46 hr/46–58 hr timepoints) wild type: 28/25/22/15, *brc-1(xoe4)* 24/43/36/9. Specific progeny counts separated by experimental replicate are presented in *Figure 1—source data 1*. (**B**) Images of wild type and *brc-1(xoe4)* mutant bivalent chromosomes displaying an absence or presence of sister chromatid exchanges (SCEs). Scale bars represent 1 µm. Dashed bordered insets contain cartoon depictions of the SCE and non-SCE bivalents that are outlined with dashed lines in the images to aid in visualizing exchange events. (**C**) Frequency of SCEs identified among wild type (n=49) or *brc-1* mutant (n=26) chromatid pairs scored. (**D–E**) Tables displaying the percent of sequenced GFP+ progeny in wild type and *brc-1* ICR assays (**D**) or non-Unc progeny IH assays (**E**) that showed signatures of mutagenic repair. Numbers in parentheses indicate the number of mutant worms out of the total number of sequenced progeny. Shaded boxes indicate timepoints in which mutant progeny were identified. The overall frequency of interhomolog assay non-Unc progeny is displayed in *Figure 1—figure supplement 3*. In panels (**D**) and (**E**), note that only mutations that allow for translation of functional GFP (panel D) or UNC-5 (panel E) protein can be detected (see Methods). In all panels, error bars represent 95% Binomial confidence intervals, dashed vertical lines delineate between timepoints within the interhomolog window (22–58 hr post heat shock) and non-interhomolog window (10–22 hr post heat shock), and p values were calculated using Fisher's Exact Test.

The online version of this article includes the following source data and figure supplement(s) for figure 1:

**Source data 1.** The source data for *Figure 1A* is provided.

**Source data 2.** The source data for *Figure 1C* is provided.

**Source data 3.** The source data for *Figure 1E* is provided.

**Figure supplement 1.** Diagram of the Intersister/Intrachromatid and Interhomolog assays.

**Figure supplement 2.** Intersister/intrachromatid repair (ICR) assay GFP+ progeny are elevated in *brc-1* and *brc-1;brd-1* mutants.

**Figure supplement 3.** Interhomolog repair is largely unperturbed in *brc-1* mutants.

**Figure supplement 4.** Illustrations of mutants identified in *brc-1* ICR and IH assays.

that could feasibly restore *unc-5* function (***Kamp et al., 2020***; ***Meier et al., 2021***). We therefore sequenced the repaired *unc-5* locus of putative noncrossover non-Unc progeny in the IH assay to confirm whether the repaired sequence matched the homolog repair template or indicated mutations at the site of Mos1 excision (see Materials and methods). Non-Unc progeny that we were unable to sequence were designated as 'undetermined non-Unc'.

When we performed the IH assay in the *brc-1* mutant, we observed a significant increase in the proportion of non-Unc progeny only at the 22–34 hr timepoint, which corresponds to the mid pachytene stage of meiosis and the end of the interhomolog window (*Figure 1—figure supplement 3A*, Fisher's Exact Test p<0.001). Mutants defective in crossover formation prolong access to the homolog as a repair template in the IH assay (*Rosu et al., 2011*), raising the possibility that *brc-1* mutants exhibit delayed prophase progression. *brc-1* mutants also exhibit delayed offloading of the early pachytene marker DSB-1 (*Trivedi et al., 2022*), and BRC-1 localizes to interhomolog recombination nodules and promotes interhomolog crossover formation (*Li et al., 2018*; *Janisiw et al., 2018*). Thus, our IH assay data is concordant with accumulating evidence that BRC-1 ensures timely meiotic prophase progression by promoting interhomolog crossover formation. However, the overall frequency of non-Unc progeny was not elevated relative to wild type within the non-interhomolog window (*Figure 1—figure supplement 3A* and 10–22 hr post heat shock, Fisher's Exact Test p=0.303), indicating that ablation of *brc-1* does not delay meiotic prophase I progression to an extent that interferes with our delineation of the interhomolog and non-interhomolog windows.

The ratio of IH assay crossover and noncrossover recombinant progeny within the interhomolog window between wild type and *brc-1* mutants was not significantly altered (*Figure 1—figure supplement 3B*, Fisher's Exact Test p=0.515). This result mirrors recombination assays previously performed in *brc-1* mutants that provided no evidence for the presence of additional crossovers (*Li et al., 2018*). Thus, our data supports a role for BRC-1 in regulating crossover recombination specifically between sister chromatids.

## BRC-1 prevents mutagenic DNA repair during the mid-pachytene stage

In both the ICR and IH assays performed in *brc-1* mutants, we identified progeny that exhibited molecular signatures of mutagenic DSB repair at the Mos1 excision site (*Figure 1D–E*, *Figure 1—figure supplement 4*). These events were only identified within the 22–34 hr timepoint, which is composed of nuclei in mid pachytene at the time of heat shock. In the ICR assay, mutants were identified as 2.4% (95% Binomial CI 0.4–12.6%) of all sequenced GFP+ progeny at the 22–34 hr time point. In the IH assay, 17.2% (95% Binomial CI 7.6–34.5%) of all sequenced non-Unc progeny at the 22–34 hr time point were identified as mutant (*Figure 1D–E*). Notably, we only sequenced GFP+ and non-Unc progeny in the ICR and IH assays, respectively. The frequency of error prone pathway utilization in a *brc-1* mutant is therefore likely much greater than our results suggest, as we could not detect mutations that disrupt the GFP or *unc-5* open reading frames.

Of the five GFP+ mutant progeny that we identified in the *brc-1* IH assay, four contained single mutations that could be attributed to meiotic repair of the Mos1-excision induced DSB (see Materials and methods), and 75% (3/4 mutations) of these meiotic mutations exhibited one or more complementary nucleotides on both ends of the deletion (*Figure 1—figure supplement 4B*). Further, the single mutant identified among *brc-1* ICR assay GFP+ progeny displayed a particularly striking duplication joined at a position sharing microhomology (*Figure 1—figure supplement 4A*). Regions of microhomology present on either end of small (<50 bp) deletions and templated insertions are characteristic of theta-mediated end joining (TMEJ) (*van Schendel et al., 2015*). A previous study demonstrated that the rate of TMEJ-mediated germline mutagenesis is elevated in *brc-1* mutants (*Kamp et al., 2020*). Our data is therefore concordant with elevated TMEJ engagement in *brc-1* mutants and further reveals that the function of BRC-1 in preventing mutagenic repair events is specifically vital in the mid-pachytene stage of meiotic prophase I.

## SMC-5/6 restricts intersister crossovers

The SMC-5/6 DNA damage complex has been hypothesized to function in homolog-independent DSB repair in *C. elegans* (*Bickel et al., 2010*). To directly assess the functions of SMC-5 in homolog-independent DSB repair, we performed the ICR assay in the *smc-5(ok2421)* null mutant. The *smc-5(ok2421)* deletion allele disrupts the final 6 exons of the 11 exons in the *smc-5* coding sequence and prevents SMC-5/6 complex assembly, as evidenced by both biochemical and cytological experiments (*Bickel et al., 2010*). Nonetheless, *smc-5(ok2421)* homozygotes develop into fertile adults. SMC-5/6 is therefore not essential for viability in *C. elegans*, unlike many other organisms (*Aragón, 2018*). Similar to *brc-1* mutants, we found that the frequency of GFP+ progeny in the ICR assay was elevated at all timepoints scored in *smc-5(ok2421)* null mutants (*Figure 2—figure supplement*

*1A*) and was not reduced relative to wild type in null *smc-6(ok3294)* mutants (*Figure 2—figure supplement 1C*). The incidence of recombinants in *smc-5* null mutants was not dramatically elevated overall in the IH assay (*Figure 2—figure supplement 2*), suggesting that Mos1 excision is not increased in *smc-5* null mutants. Since we cannot be certain of the cause of this increase in recombinants in the *smc-5* ICR assay, we refrain from making conclusions regarding the absolute frequency of intersister repair events that occur within this mutant context. Regardless of the absolute number of ICR assay GFP+ recombinant progeny, we did identify both crossover and noncrossover recombinants at all timepoints scored in our *smc-5* ICR assays, demonstrating that SMC-5/6 alone is not required for noncrossover nor crossover homolog-independent repair (*Figure 2—figure supplement 1A*).

To determine if SMC-5 regulates intersister/intrachromatid recombination outcomes, we examined occurrence of *smc-5* ICR assay crossover recombinants as a proportion of all recombinants identified. While the proportion of crossovers was not significantly different than wild-type within the individual 12 hr timepoints we scored (*Figure 2A*), the frequency of crossover recombinants in *smc-5* mutants was significantly elevated within the interhomolog window overall (*Figure 2B*, Fisher's Exact Test p=0.037). Thus, our data suggests that a function of SMC-5 is to prevent homolog-independent crossovers arising from DSBs induced in early stages of meiotic prophase I. To cytologically affirm that intersister crossovers occur in *smc-5* mutants, we assessed the frequency of SCEs in *smc-5(ok2421)* mutants by examining EdU labeled chromatids at diakinesis. Since mutants of *smc-5* are known to have defects in chromosome compaction and produce misshapen bivalents (*Bickel et al., 2010*; *Hong et al., 2016*), the majority of chromatid pairs in the *smc-5* null mutant were uninterpretable in the EdU labeling assay (See Methods). Despite these challenges, we identified SCEs in 35.7% of scored chromatid pairs (*Figure 2C–D*, 5/14 chromatid pairs scored, 95% Binomial CI 16.3–61.2%, Fisher's Exact Test p=0.004) as compared to only 4.1% of wild type chromatid pairs (2/49 chromatid pairs scored, 95% Binomial CI 1.1–13.7%) (*Almanzar et al., 2021*). While we do not consider these results as an absolute metric for the rate of SCEs in *smc-5* mutants due to limited sample size from chromosome morphology issues, our ICR assay and EdU labeling experiments nevertheless demonstrate that intersister repair still occurs in the absence of the fully assembled SMC-5/6 complex and indicate a role for SMC-5 in regulating intersister crossing over during *C. elegans* meiosis.

## SMC-5/6 is not required for efficient interhomolog recombination

To determine if SMC-5 regulates interhomolog recombination, we performed the IH assay in the *smc-5(ok2421)* null mutant. We identified both interhomolog crossover and noncrossover recombinants in the IH assay (*Figure 2—figure supplement 2*), indicating that SMC-5/6 is not required for either of these recombination pathways. Similar to *brc-1* mutants, we noted elevated non-Unc progeny at the 22–34 hr time point in *smc-5* mutants, implying that meiotic prophase progression may be slightly delayed when SMC-5/6 function is lost (*Figure 2—figure supplement 2A*, Fisher's Exact Test p<0.001). Importantly, non-Unc progeny were not increased in the non-interhomolog window in *smc-5* mutants (*Figure 2—figure supplement 2A*, Fisher's Exact Test p=1.000), so any delay in meiotic prophase I that may be caused by *smc-5* loss of function did not interfere with our delineation of the interhomolog the non-interhomolog windows. The proportion of interhomolog crossover recombinants among all recombinants identified also was not altered in an *smc-5* mutant (*Figure 2—figure supplement 2B*, Fisher's Exact Test p=0.495). Thus, the IH assay data does not provide evidence that SMC-5/6 has a major role in ensureing efficient interhomolog recombination.

Among all sequenced ICR and IH assay GFP+ and non-Unc progeny isolated in *smc-5* mutants, we identified only one mutagenic DSB repair event at the 22–34 hr timepoint of the IH assay (*Figure 2E*, *Figure 2—figure supplements 1A and 2A*). Moreover, the frequency of *smc-5* non-Unc mutants that we detected at this timepoint (1.32% of all sequenced non-Unc progeny, 95% Binomial CI 0.2–7.1%) is lower than the frequency observed in *brc-1* mutants (Fisher's Exact Test p=0.015). Profiling of meiotic mutagenic DNA repair events in *smc-6* mutants has indicated that large structural variations are a primary class of mutations that arise in SMC-5/6 deficient germlines (*Volkova et al., 2020*). Thus, it is likely that a greater frequency of DSBs were repaired by mutagenic pathways in the *smc-5* ICR and IH assays than we detected; if these products disrupted the coding sequence of GFP or *unc-5* respectively, however, then they would have escaped detection in our assays.

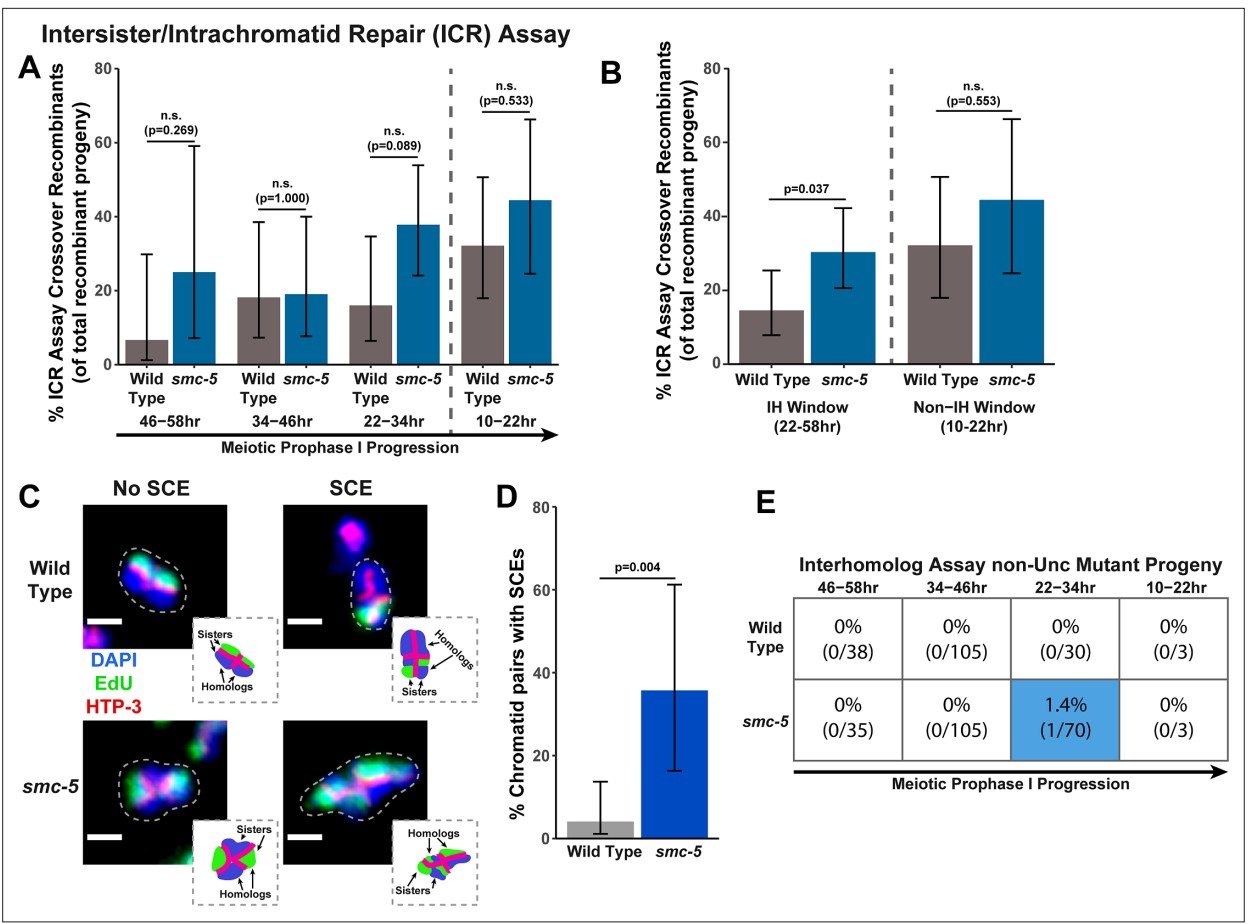

**Figure 2.** SMC-5 represses intersister crossovers. (**A**) Bar plot displaying the percent of crossover recombinant progeny identified in wild type and *smc-5* ICR assays out of all recombinant progeny scored within individual 12 hr timepoint periods. Frequencies of recombinants identified overall in ICR assays is displayed in *Figure 2—figure supplement 1*. Wild type ICR assay data is shared in *Figures 1A, 2A, B and 4A*, *Figure 1—figure supplement 2*, *Figure 2—figure supplement 1*, and *Figure 4—figure supplement 1*. The total number of progeny scored in each dataset are (10–22 hr/22–34 hr/34–46 hr/46–58 hr timepoints) wild type: 28/25/22/15, *smc-5(ok2421)* 18/37/21/8. Specific progeny counts separated by experimental replicate are presented in *Figure 2—source data 1*. (**B**) Bar plot displaying the percent of crossover recombinant progeny identified in wild type and *smc-5* ICR assays out of all recombinant progeny scored within the interhomolog window (22–58 hr post heat shock) and non-interhomolog window (10–22 hr post heat shock). Data is shared with panel (**A**). (**C**) Images of wild type and *smc-5(ok2421)* mutant bivalent chromosomes displaying an absence or presence of SCEs. Scale bars represent 1 μm. Dashed bordered insets contain cartoon depictions of the SCE and non-SCE bivalents that are outlined with dashed lines in the images to aid in visualizing exchange events. (**D**) Frequency of SCEs identified among wild type (n=49) or *smc-5(ok2421)* mutant (n=14) chromatid pairs scored. (**E**) Table displaying the percent of sequenced non-Unc progeny in wild type and *smc-5* IH assays that showed signatures of mutagenic repair. Numbers in parentheses indicate the number of mutant worms out of the total number of sequenced progeny. Colored boxes indicate timepoints in which mutant progeny were identified. The overall frequency of interhomolog assay non-Unc progeny is displayed in *Figure 2—figure supplement 2*. Note that only mutations that allow for translation of functional UNC-5 protein can be detected (see Methods). In all panels, error bars represent 95% Binomial confidence intervals, dashed vertical lines delineate between timepoints within the interhomolog window (22–58 hr post heat shock) and non-interhomolog window (10–22 hr post heat shock), and p values were calculated using Fisher's Exact Test.

The online version of this article includes the following source data and figure supplement(s) for figure 2:

**Source data 1.** The source data for *Figure 2A* is provided.

**Source data 2.** The source data for *Figure 2D* is provided.

**Source data 3.** The source data for *Figure 2E* is provided.

**Figure supplement 1.** Intersister/intrachromatid repair (ICR) assay GFP+ progeny are elevated in *smc-5* mutants.

**Figure supplement 2.** Interhomolog repair is largely unperturbed in *smc-5* mutants.

**Figure supplement 3.** Illustrations of mutants identified in *smc-5* IH assays.

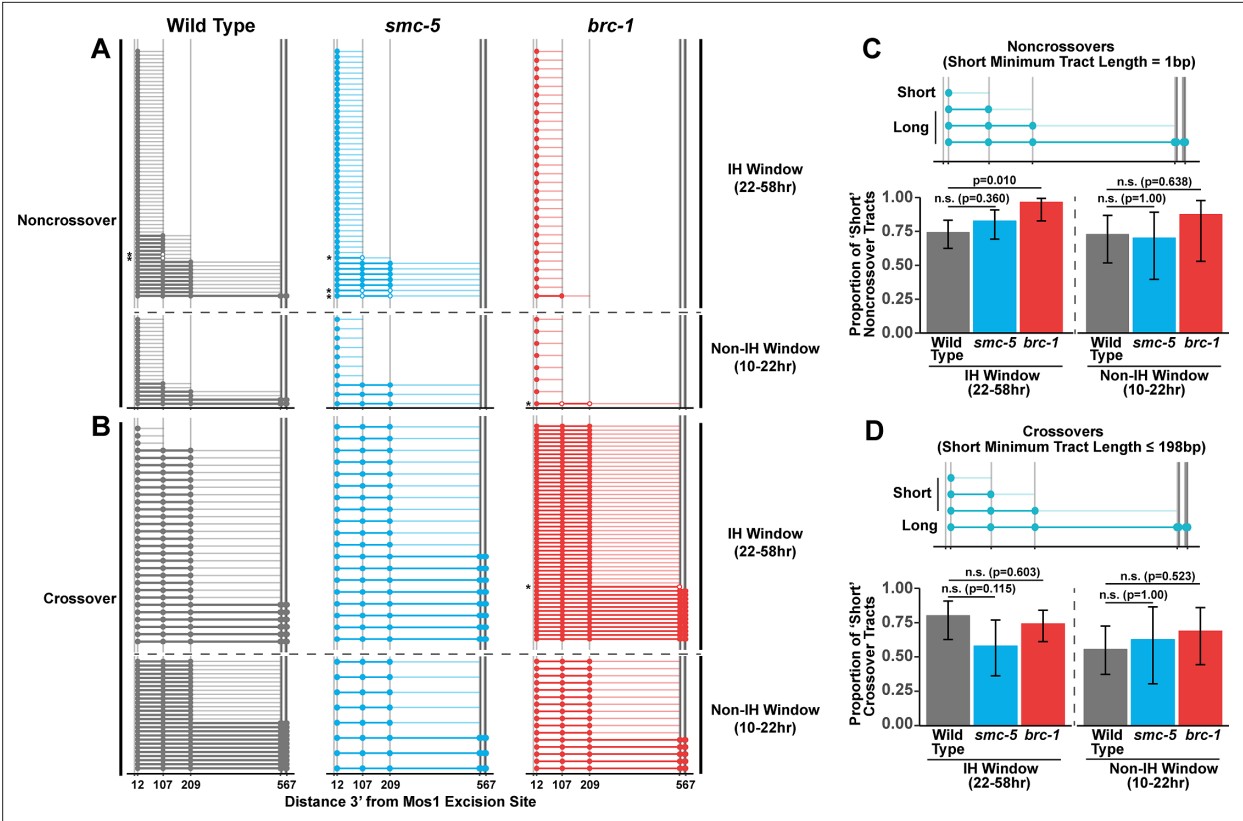

**Figure 3.** BRC-1 is required for long noncrossover gene conversion. (**A–B**) Plots of conversion tracts sequenced from recombinant ICR assay loci. Vertical grey lines indicate the positions of polymorphisms in the ICR assay with bp measurements given 3' relative to the site of Mos1 excision (*Toraason et al., 2021b*; *Toraason et al., 2021a*;). Each horizontal line represents a single recombinant sequenced, ordered from smallest tract to largest tract within the interhomolog and non-interhomolog windows. Filled in points represent fully converted polymorphisms, while points with white interiors represent heteroduplex DNA sequences identified in conversion tracts. Tracts containing heteroduplex are marked with asterisks. High opacity horizontal lines within plots represent the minimum conversion tract length, or the distance from the most proximal to the most distal converted polymorphisms. Low opacity horizontal lines indicate the maximum conversion tract, extending from the most distal converted polymorphism to its most proximal unconverted polymorphism. Tracts from noncrossover recombinants are displayed in A, while tracts from crossover recombinants are displayed in (**B**). (**C–D**) Frequency of short noncrossover tracts (C, minimum tract length 1 bp converted at only the 12 bp polymorphism) or short crossover tracts (D, minimum tract length 198 bp) as a proportion of all tracts identified from progeny laid within the interhomolog and non-interhomolog windows. Error bars represent the 95% binomial confidence intervals of these proportions and p values were calculated using Fisher's Exact Test. Diagrams above bar plots depict the sizes of tracts considered 'long' or 'short' in each respective group. In all panels, dashed grey lines delineate between the interhomolog window (22–58 hr post heat shock) and non-interhomolog window (10–22 hr post heat shock) timepoints.

The online version of this article includes the following source data for figure 3:

**Source data 1.** The source data for *Figure 3* is provided.

## BRC-1 promotes the formation of long homolog-independent noncrossover conversion tracts

Since we identified functions for BRC-1, and to a lesser extent SMC-5, in restricting intersister crossover recombination, we wanted to determine if recombination intermediate processing is altered in *brc-1* and *smc-5* mutants. Evaluation of sequence conversions have informed much of our understanding of recombination intermediate processing (*Pâques and Haber, 1999*; *Szostak et al., 1983*; *Ahuja et al., 2021*; *Marsolier-Kergoat et al., 2018*; *Cole et al., 2010*; *Cole et al., 2014*; *Medhi et al., 2016*; *Hum and Jinks-Robertson, 2017*). The ICR assay was engineered to contain multiple polymorphisms spanning 12 bp to 567bp 3' from the site of Mos1 excision, enabling conversion tract analysis of homolog-independent recombination (*Toraason et al., 2021c*). In a wild type context, 74.2% of ICR assay noncrossover conversion tracts within the interhomolog window are 'short', which we define as tracts with a sequence conversion only at the most proximal polymorphism 12 bp downstream from the site of Mos1 excision (*Figure 3A and C*, wild type 74.2% short tracts 95% CI 62.6–83.3%).

In contrast, 96.6% of *brc-1* noncrossover tracts during the interhomolog window were 'short' (*brc-1* interhomolog window 96.6% short tracts 95% CI 82.8–99.4%, Fisher's Exact Test p=0.010). During the non-interhomolog window, we could not detect an effect of null mutation of *brc-1* on the proportion of 'short' noncrossover tracts (*Figure 3A and C*, wild type 72.7% short tracts 95% CI 51.8–86.8%; *brc-1* 87.5% short tracts 95% CI 52.9–97.8%, Fisher's Exact Test p=0.638). However, due to the low frequency of noncrossover events in *brc-1* mutants within the single non-IH window timepoint, our dataset does not exclude the possibility that BRC-1 may also influence the mechanisms of noncrossover formation in later stages of prophase I.

We previously showed that wild type intersister/intrachromatid crossover conversion tracts in *C. elegans* tend to be larger than noncrossovers, with a median minimum conversion tract length (the distance from the most proximal to the most distal converted polymorphisms in bp) for intersister/intrachromatid crossovers being 198 bp (*Figure 3B*; *Toraason et al., 2021c*). Based on this median length for intersister/intrachromatid crossovers, we defined 'short' ICR assay crossover tracts as ≤198 bp in length. We found that the proportion of 'short' crossover tracts was not altered by *brc-1* mutation within the interhomolog window (*Figure 3B and D*, wild type 80.0% short tracts 95% CI 62.7–90.5%; *brc-1* 74.1% short tracts 95% CI 61.1–83.9%, Fisher's Exact Test *P*=1.000) nor within the non-interhomolog window (*Figure 3B and D*, wild type 55.6% short tracts 95% CI 37.3–72.4%; *brc-1* 68.8% short tracts 95% CI 44.4–58.8%, Fisher's Exact Test p=0.657). Taken together, these results support a model in which BRC-1 regulates mechanisms of intersister/intrachromatid recombination that determine noncrossover gene conversion tract length (but not crossover tract length) during meiotic prophase I (see Discussion).

## SMC-5 does not regulate the extent of homolog-independent gene conversion

To assess if SMC-5/6 influences recombination intermediates, we compared *smc-5* mutant ICR assay conversion tracts to their wild type counterparts. We found that ICR assay noncrossover conversion tracts in *smc-5* mutants exhibited a similar proportion of 'short' tracts to wild type in both the interhomolog (*Figure 3A and C*, wild type 74.2% short tracts 95% CI 62.6–83.3%; *smc-5* 82.6% short tracts 95% CI 69.3–90.9%, Fisher's Exact Test p=0.360) and non-interhomolog windows (*Figure 3A and C*, wild type 72.7% short tracts 95% CI 51.8–86.8%; *smc-5* 70% short tracts 95% CI 39.7–89.2%). Thus, SMC-5/6 does not have a strong effect on the extent of noncrossover gene conversion in intersister/intrachromatid repair.

When we compared the proportion of 'short' *smc-5* ICR assay crossover tracts to wild type, we similarly observed that there is no significant difference in the proportion of short and long crossover tracts in either the interhomolog (*Figure 3B and D*, wild type 80.0% short tracts 95% CI 62.7–90.5%; *smc-5* 57.9% short tracts 95% CI 36.3—76.9%, Fisher's Exact Test p=1.000) or non-interhomolog windows (*Figure 3B and D*, wild type 55.6% short tracts 95% CI 37.3–72.4%; *smc-5* 62.5% short tracts 95% CI 30.6–86.3%, Fisher's Exact Test p=1.000). Taken together, these results do not support a function for SMC-5/6 in regulating the extent of noncrossover and crossover gene conversion that yields functional GFP repair products.

In our wild type, *brc-1*, and *smc-5* ICR assay conversion tracts, we additionally noted multiple instances of heteroduplex DNA in our sequencing (*Figure 3A and B* asterisks). DNA heteroduplex is a normal intermediate when recombination occurs between polymorphic templates but is usually resolved by the mismatch repair machinery (*Ahuja et al., 2021*; *Guo et al., 2017*). Our observation of these events across genotypes suggests that at a low frequency, mismatch repair may fail to resolve heteroduplex DNA during the course *C. elegans* meiotic DSB repair.

## Depletion of SMC-5 does not enhance the homolog-independent recombination defects of *brc-1* mutants

SMC-5/6 and BRC-1 genetically interact to regulate multiple processes that influence meiotic genomic fidelity, including mitotic DNA replication and chromosome compaction (*Hong et al., 2016*; *Wolters et al., 2014*). To test if SMC-5/6 and BRC-1 genetically interact to regulate homolog-independent DNA repair outcomes, we performed the ICR assay in a background where both BRC-1 and SMC-5 are absent. Since the genetic instability and sterility of *smc-5;brc-1* double mutants precluded the use the ICR assay in this particular genetic background, we employed the auxin-inducible degron

(AID) system (*Zhang et al., 2015*) to conditionally deplete SMC-5 in the germline of *brc-1* mutants (*smc-5(syb3065[AID\*::3xFLAG]);brc-1(xoe4)*). Importantly, the addition of a C-terminal AID\* tag to the *smc-5* coding sequence did not convey germline sensitivity to exogenous DSBs except in the presence of auxin (*Figure 4—figure supplement 2A–B*), indicating that this tag does not severely compromise the function of SMC-5 until conditional depletion. When we performed the ICR assay in *smc-5::AID\*::3xFLAG;brc-1* mutants following SMC-5 germline depletion, we identified GFP+ noncrossover and crossover ICR assay recombinant progeny in *smc-5::AID\*;brc-1* mutants (*Figure 4—figure supplement 1*), reinforcing that neither of these repair outcomes require SMC-5 or BRC-1. Further, the proportion of ICR assay crossover recombinants in *smc-5::AID\*;brc-1* mutants was elevated relative to wild type at both the 22–34 hr and 46–58 hr timepoints within the interhomolog window but was indistinguishable from *brc-1* single mutants at all timepoints assayed (*Figure 4A*). Overall, this result suggests that depletion of SMC-5 does not grossly disrupt homolog-independent repair in *brc-1* mutants and is consistent with a model in which BRC-1 acts in the same pathway with SMC-5/6 in regulating intersister repair outcomes.

## BRC-1 promotes recombinase loading to DSBs in *smc-5* mutants

To determine the interactions of SMC-5 and BRC-1 in regulating early steps of meiotic DSB repair, we examined the localization of the recombinase RAD-51, which marks DSBs undergoing early steps of recombination (*Colaiácovo et al., 2003*), in *smc-5*, *brc-1*, and *smc-5;brc-1* mutants (*Figure 4B*). The number of RAD-51 foci per nucleus are reduced at early-mid pachytene in *brc-1* null mutants relative to wild type due to ectopic engagement of error-prone pathways that antagonize the formation of RAD-51 intermediates (*Li et al., 2018*; *Janisiw et al., 2018*; *Li et al., 2020*; *Trivedi et al., 2022*). In contrast, mutants for either *smc-5* or *smc-6* exhibit highly elevated levels of RAD-51 throughout prophase I (*Bickel et al., 2010*). Our results corroborate these previous studies, as we observed that *brc-1* mutants exhibit reduced RAD-51 foci relative to wild type in early through mid pachytene (*Figure 4B*) and that *smc-5* mutants exhibited elevated RAD-51 foci compared to wild type at all stages except late pachytene (*Figure 4B*). Strikingly, we observed that the increased number of RAD-51 foci per nucleus in *smc-5* mutants was suppressed when *brc-1* was also mutated (*Figure 4B*). This result is reminiscent of interactions between BRC-1 and SMC-5/6 in the mitotic germline, as BRC-1 is required for localization of RAD-51 to collapsed replication forks that arise in *smc-5* mutants (*Wolters et al., 2014*). BRC-1 is required to load and/or maintain RAD-51 localization to DSBs at mid/late pachytene when interhomolog recombination is perturbed (*Li et al., 2018*; *Janisiw et al., 2018*) as well as following induction of exogenous DSBs from sources such as ionizing radiation (*Figure 4—figure supplement 4C*; *Li et al., 2018*; *Janisiw et al., 2018*). Our results expand upon these prior analyses and identify functions for BRC-1 in promoting recombination repair of DSBs throughout meiotic prophase I in the absence of the SMC-5/6 complex.

## BRC-1 sensitizes *smc-5* mutants to exogenous DSBs by restricting error-prone repair pathways

The genetic instability and defects in RAD-51 localization that we observed in *smc-5;brc-1* mutants contrasted with the lack of a strong phenotype in ICR assay outcomes when SMC-5 was depleted in *brc-1* mutants. This incongruity raised the possibility that SMC-5 and BRC-1 may interact to regulate recombination-independent DNA repair pathways. To determine the functional interplay of SMC-5/6 and BRC-1 in regulating meiotic DSB repair, we assessed the sensitivity of *smc-5*, *brc-1*, and *smc-5;brc-1* mutant gametes to exogenous DSBs induced by ionizing radiation. Accordingly, we treated adult hermaphrodites of each genotype with 0, 2500, or 5000 Rads of ionizing radiation, which induces an estimated 9.8 and 19.6 DSBs/chromosome pair, respectively (*Yokoo et al., 2012*), and assayed the resultant progeny derived from their irradiated oocytes for larval viability (*Figure 4C*) during a similar reverse time course as was done in our ICR and IH assays (see Materials and methods), thereby enabling us to identify meiosis-stage-specific DNA repair defects in these mutants.

We noted differences in the brood viabilities of individual genotypes and variation between hermaphrodites within those genotypes even in unirradiated conditions (*Figure 4C*). These baseline disparities posed a challenge in interpreting the effects of ionizing radiation on brood viability, as the resilience of an irradiated cohort will be affected by both underlying fertility defects as well as the effects of the exogenous DNA damage that we sought to quantify. To estimate the effect of

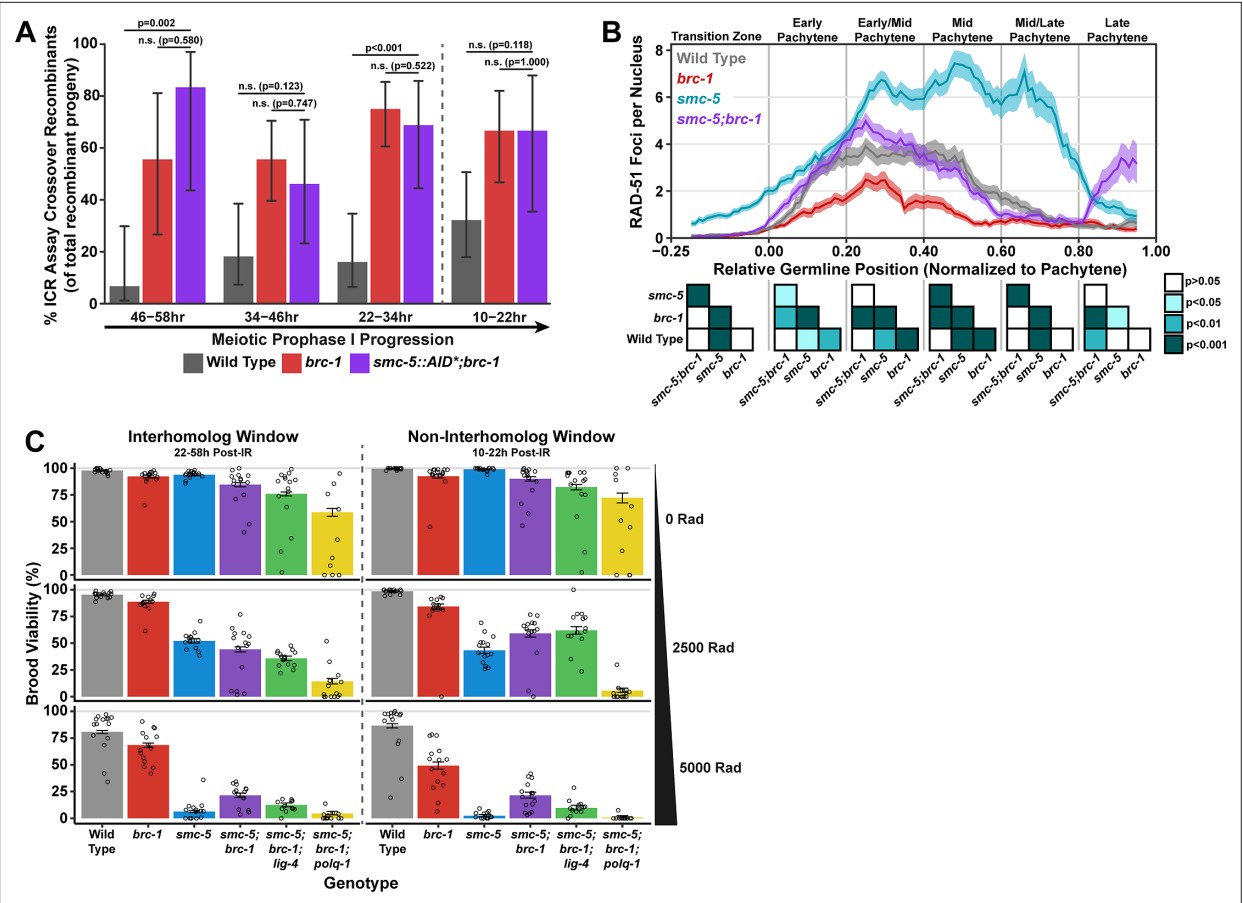

**Figure 4.** Interactions of SMC-5/6 and BRC-1 in meiotic DSB repair. (**A**) Bar plot displaying the percent of crossover recombinant progeny identified in wild type, *brc-1*, and *smc-5::AID\*::3xFLAG;brc-1* ICR assays out of all recombinant progeny scored within individual 12 hr timepoint periods. Frequencies of recombinants identified overall in ICR assays is displayed in *Figure 4—figure supplement 1*. Wild type ICR assay data is shared in *Figures 1A, 2A, B and 4A*, *Figure 1—figure supplement 2*, *Figure 2—figure supplement 1*, and *Figure 4—figure supplement 1*. *brc-1* ICR assay data is shared in *Figure 1A*, *Figure 1—figure supplement 2*, and *Figure 4—figure supplement 1*. The total number of recombinant progeny scored in each dataset are (10–22 hr/22–34 hr/34–46 hr/46–58 hr timepoints) wild type: 28/25/22/15, *brc-1(xoe4)* 24/43/36/9, *smc-5::AID\*::3xFLAG;brc-1* 9/16/13/6. Specific progeny counts separated by experimental replicate are presented in *Figure 4—source data 3*. (**B**) RAD-51 foci per nucleus displayed as a sliding window (window width 0.1 position units, step size 0.01 position units) across the length of the germline with germline position normalized to the length of pachytene (see Methods). Lines represent the mean RAD-51 foci per nucleus while the shaded area represents SEM within a given window. RAD-51 foci were compared between genotypes within six bins across the germline divided by germline distance: transition zone (−0.25–0), early pachytene (0–0.2), early/mid pachytene (0.2–0.4), mid pachytene (0.4–0.6), mid/late pachytene (0.6–0.8), and late pachytene (0.8–0.1). Heat maps below each bin display the p values for pairwise Mann-Whitney U tests with Holm-Bonferroni correction for multiple comparisons within each bin. For each genotype, n≥7 germlines derived from at least three experimental replicates were analyzed and combined. The number of nuclei scored in each bin (TZ, EP, E/MP, MP, M/LP, LP) are: wild type 258/137/135/121/114/95, *smc-5* 402/395/348/285/217/189, *brc-1* 231/170/161/186/159/121, *smc-5;brc-1* 202/182/167/157/122/82. The per-nucleus RAD-51 counts from this analysis are available in *Figure 4—source data 2*. (**C**) Brood viabilities of irradiated hermaphrodites. Bar plots represent the population brood viability and error bars indicate the 95% Binomial confidence interval of this value. The brood viabilities of individual hermaphrodites scored are indicated by data points. For all genotypes and radiation treatments presented in this panel, three experimental replicates were performed with n=5 hermaphrodites scored per condition in each replicate for a total of n=15 hermaphrodite broods scored for each genotype and condition combination. Individual hermaphrodite progeny counts are presented in *Figure 4—source data 3*. Statistical analysis of the data presented in this panel is depicted in *Figure 4—figure supplement 3*. Localization of BRC-1::GFP and SMC-5::AID\*::3xFLAG in respective *smc-5* and *brc-1* mutants and in the presence of ionizing radiation is presented in *Figure 4—figure supplements 4 and 5*. A proposed molecular model for the interactions between BRC-1 and SMC-5/6 is presented in *Figure 4—figure supplement 6*.

The online version of this article includes the following source data, source code, and figure supplement(s) for figure 4:

**Source data 1.** The source data for *Figure 4A* is provided.

**Source data 2.** The source data for *Figure 4B* is provided.

**Source data 3.** The source data for *Figure 4C* is provided.

*Figure 4 continued on next page*

*Figure 4 continued*

**Figure supplement 1.** Intersister/intrachromatid repair (ICR) assay defects in *brc-1* mutants are not exacerbated by SMC-5 depletion.

**Figure supplement 2.** Loss of BRC-1, but not endogenous tagging of SMC-5::AID*::3xFLAG, inhibits RAD-51 localization to irradiation-induced DSBs.

**Figure supplement 2—source data 1.** The source data for *Figure 4—figure supplement 2B* is provided.

**Figure supplement 3.** Bayesian analysis of brood viability following ionizing radiation treatment.

**Figure supplement 3—source data 1.** The source data for *Figure 4—figure supplement 3A* is provided.

**Figure supplement 3—source code 1.** The source code for *Figure 4-figure supplement 3A and 3C* is provided.

**Figure supplement 4.** SMC-5/6 is not required for GFP::BRC-1 localization.

**Figure supplement 5.** BRC-1 is not required for SMC-5::AID*::3xFLAG localization.

**Figure supplement 6.** Proposed model of BRC-1 and SMC-5/6 function in *C. elegans* intersister DSB repair.

ionizing radiation on brood viability and to account for inter-hermaphrodite variance in our analysis, we employed a Bayesian hierarchical statistical modeling approach using our dataset (*Figure 4—figure supplement 3*, see Materials and methods). From this analysis, we calculated a metric termed 'gamma' for each genotype, representing the sensitivity of a given genotype to ionizing radiation (*Figure 4—figure supplement 3A*, see Materials and methods). A gamma estimate of 1 indicates that irradiation has no effect on brood viability, while a gamma estimate of 0 indicates that all progeny of a genotype are inviable following irradiation.

To assess the differential sensitivities of *smc-5*, *brc-1*, and *smc-5;brc-1* mutants across meiotic prophase I, we compared the 95% credible intervals of the gamma estimates for each genotype within the interhomolog and the non-interhomolog windows for both moderate (2500 Rads) and high (5000 Rads) irradiation doses (*Figure 4—figure supplement 3A*). Across all irradiation doses and timepoints, we note that loss of *smc-5* conveys a greater sensitivity to exogenous DNA damage than loss of *brc-1* (*Figure 4C*, *Figure 4—figure supplement 3A*), emphasizing that the SMC-5/6 complex prevents catastrophic defects following exogenous DNA damage induction. The sensitivity of *brc-1* mutants was greater in the non-interhomolog window at high doses of irradiation (*Figure 4C*, *Figure 4—figure supplement 3A*), suggesting that nuclei in late pachytene become more dependent upon BRC-1 to efficiently resolve DNA damage.

At 2500 Rad of ionizing radiation, we found that mutation of both *smc-5* and *brc-1* differentially impacted radiation resilience within the interhomolog and non-interhomolog windows. In the interhomolog window, the *smc-5;brc-1* double mutant and *smc-5* single mutant gamma estimates overlap, indicating that loss of BRC-1 does not alter *smc-5* mutant sensitivity at this timepoint (*Figure 4C*, *Figure 4—figure supplement 3A*). Further, *brc-1* mutant gamma estimates are indistinguishable from wild type within the interhomolog window (*Figure 4C*, *Figure 4—figure supplement 3A*); therefore, the absence of an interaction may reflect the dispensability of BRC-1 in early prophase I for progeny survival when DNA damage levels are not extreme. In the non-interhomolog window, however, we observed a striking suppression of sensitivity to exogenous DSBs in *smc-5;brc-1* double mutants as compared to *smc-5* single mutants (*Figure 4C*, *Figure 4—figure supplement 3A*). This synthetic suppression of sensitivity is recapitulated across meiotic prophase I at 5000 Rads of ionizing radiation in *smc-5;brc-1* double mutants (*Figure 4C*, *Figure 4—figure supplement 3A*). Thus, our data indicates that DNA damage sensitivity observed in *smc-5* mutants is enhanced by BRC-1-mediated functions.

The synthetic suppression of radiation-induced brood lethality and reduced RAD-51 foci in *smc-5;brc-1* double mutants prompted us to hypothesize that recombination-independent pathways may be engaged to repair DSBs when both SMC-5 and BRC-1 are ablated, thereby alleviating the DSB repair defects of *smc-5* mutants. BRC-1 is known to repress both TMEJ and NHEJ in multiple organisms, including *C. elegans* (*Kamp et al., 2020*; *Li et al., 2020*; *Huen et al., 2010*), and ablation of NHEJ or TMEJ is sufficient to partially rescue RAD-51 foci loading to DSBs in *brc-1* mutants (*Trivedi et al., 2022*). To test whether TMEJ and/or NHEJ contribute to the ionizing radiation resilience observed in *smc-5;brc-1* double mutants, we created *smc-5;brc-1;polq-1* and *smc-5;brc-1;lig-4* triple mutants that are defective in TMEJ and NHEJ respectively. *lig-4* mutation did not fully suppress the synthetic radiation resilience of *smc-5;brc-1* mutants, suggesting that NHEJ is not a primary mechanism of DNA repair in meiotic nuclei when both SMC-5/6 and BRC-1 are lost. However, we observed a

striking effect in *smc-5;brc-1;polq-1* mutants; even at the moderate dose of 2500 Rads, loss of POLQ-1 caused dramatic sensitization of *smc-5;brc-1* mutants to ionizing radiation (*Figure 4C*, *Figure 4—figure supplement 3A*). This effect was particularly strong in the non-interhomolog window, where *smc-5;brc-1;polq-1* mutants were nearly sterile following ionizing radiation treatment regardless of irradiation dose (*Figure 4C*, *Figure 4—figure supplement 3A*). These results strongly indicate that *smc-5;brc-1* deficient germ cells exposed to exogenous DNA damage are dependent upon TMEJ for fertility.

Since irradiation of adult hermaphrodites also induces additional DSBs in mature sperm, we wanted to ensure impacts on brood viability were not due to sperm-specific defects. To test the contributions of sperm effects to our radiation sensitivity analysis, we determined the viability of oocytes from irradiated hermaphrodites mated to unirradiated wild type males (*Figure 4—figure supplement 3B*). DSBs induced in sperm are primarily repaired by TMEJ in the fertilized zygote and have minimal impact on F1 viability when POLQ-1 is present (*Wang et al., 2023*), so we focused our analysis on *polq-1*, *smc-5;brc-1*, and *smc-5;brc-1;polq-1* null mutants. As has been previously reported (*Wang et al., 2023*), we found that at high doses of radiation *polq-1* mutants exhibited a sperm-specific fertility defect (*Figure 4—figure supplement 3A–B*). A similar restoration of viability was observed in the interhomolog window of *smc-5;brc-1;polq-1* mutants, indicating that the effect of *polq-1* on oocytes irradiated in early stages of prophase I was primarily due to sperm-specific effects. However, the increased radiation sensitivity of *smc-5;brc-1;polq-1* within the non-interhomolog window was not rescued by mating (*Figure 4—figure supplement 3A–B*). Thus, our data indicate that *smc-5;brc-1* oocytes become dependent upon TMEJ for DSB repair in late stages of meiotic prophase I.

Taken together, the results of our irradiation analysis indicate that both SMC-5/6 and BRC-1 contribute to gamete viability following ionizing radiation treatment, with loss of SMC-5/6 having far greater consequences for the gamete than loss of BRC-1 (*Figure 4C*, *Figure 4—figure supplement 3A*). As *brc-1* mutation confers synthetic resilience to radiation in *smc-5* mutants, we provide evidence that some functions of BRC-1 contribute to the meiotic DSB repair defects associated with loss of *smc-5*. Further, we find that TMEJ is vital to radiation resilience in late meiotic prophase I in *smc-5;brc-1* mutants, suggesting that this pathway compensates for the DNA repair deficiencies incurred when SMC-5/6 and BRC-1 are both lost (*Figure 4—figure supplement 3D*). Repression of TMEJ by BRC-1 may therefore be deleterious to reproductive success in *smc-5* null mutants by enabling more severe DNA repair errors to occur.

## BRC-1 localization is independent of SMC-5/6

To determine whether there is a co-dependency between BRC-1 and SMC-5/6 for localization, we first examined GFP::BRC-1 by immunofluorescence in both wild type and *smc-5* mutant germlines. Similar to previous studies (*Li et al., 2018*; *Janisiw et al., 2018*), we observed that BRC-1 localizes as a nuclear haze in the premeiotic tip through early pachytene and becomes associated with the synaptonemal complex during the progression of pachytene in wild-type germlines (*Figure 4—figure supplement 4*). In late pachytene, BRC-1 relocates to the short arms of the bivalents, where it can be visualized at diplotene as short tracks on the compacted chromosome arms (*Figure 4—figure supplement 4*). When we examined *smc-5* mutants, the general pattern of GFP::BRC-1 localization across meiotic prophase was similar to wild type, except in the premeiotic tip where GFP::BRC-1 formed bright foci (*Figure 4—figure supplement 4*). Given that BRD-1, the obligate heterodimeric partner of BRC-1, was found to form a similar localization in *smc-5* mutants (*Wolters et al., 2014*), the bright GFP::BRC-1 foci in the pre-meiotic tip likely mark BRC-1 localization to collapsed replication forks (*Bickel et al., 2010*; *Wolters et al., 2014*). Our data therefore indicate that the localization of BRC-1 does not require SMC-5/6.

To assess if BRC-1 changes localization in response to exogenous DSBs, we exposed wild type and *smc-5* mutant germlines to 5000 Rads of ionizing radiation and again examined germline GFP::BRC-1 by immunofluorescence. We found that the general pattern of GFP::BRC-1 localization was not altered following irradiation in both wild type and *smc-5* mutants (*Figure 4—figure supplement 4*). Taken together, our results suggest that BRC-1 localization is not altered following the induction of exogenous DSBs even when SMC-5/6 complex function is lost.

## SMC-5/6 localization is independent of BRC-1

To determine whether SMC-5/6 localization is dependent upon BRC-1, we examined the localization of SMC-5::AID*::3xFLAG in both wild type and *brc-1* mutants (*Figure 4—figure supplement 5*). We observed that, similar to a prior study (*Bickel et al., 2010*), SMC-5/6 is present in meiotic nuclei within meiotic prophase I. Notably, we found that SMC-5 staining in early and mid-pachytene was primarily localized to the chromosome axis, marked with HTP-3 (*Figure 4—figure supplement 5*). This localization pattern was altered in the transition to the diplotene stage, when we observed that SMC-5 localizes to the chromatin on the compacting bivalent chromosomes, matching previous analysis (*Figure 4—figure supplement 5*; *Bickel et al., 2010*). The pattern of SMC-5 localization was not disrupted in a *brc-1* mutant, and similarly was not altered following exposure to 5000 Rads of ionizing radiation (*Figure 4—figure supplement 5*). Thus, the localization of SMC-5/6 does not depend upon the activity of BRC-1 and is not altered following induction of exogenous DNA damage at the levels we tested.

## Discussion

Meiotic cells must coordinate DNA repair pathway engagement to ensure both formation of interhomolog crossovers and repair of all DSBs. The highly conserved proteins SMC-5/6 and BRC-1 promote accurate DSB repair, but the specific DNA repair outcomes that these proteins regulate have remained unclear. We find that SMC-5/6 and BRC-1 act to repress intersister crossovers and further demonstrate that BRC-1 specifically promotes long homolog-independent noncrossover gene conversion. We also observe that mutants for *brc-1* incur DNA repair defects at mid pachytene, as evidenced by increased engagement of error prone repair pathways and deficiencies in RAD-51 localization to DSBs. By comparing the germ cell resilience of *smc-5*, *brc-1,* and *smc-5;brc-1* mutants to ionizing radiation, we reveal that BRC-1 enhances the meiotic DNA repair defects of *smc-5* mutants and provide evidence that this interaction is in part underpinned by BRC-1 dependent repression of TMEJ and promotion of recombination. Taken together, our study illuminates specific functions and interactions of highly conserved DNA repair complexes in promoting germline genome integrity.

### BRC-1 and SMC-5/6 as 'intersister repair' proteins

Foundational work defining the phenotypes of *brc-1* and *smc-5/6* loss-of-function mutants provided rationale that these proteins regulate homolog-independent DSB repair (*Bickel et al., 2010*; *Adamo et al., 2008*). Our study utilizes genetic and cytological technologies not available at the time of this initial work to specifically examine homolog-independent repair in the absence of *brc-1* and *smc-5*. While we find that neither BRC-1 nor SMC-5/6 are necessary for intersister repair in *C. elegans* meiosis per se, we instead identified functions for both BRC-1 and SMC-5 in regulating intersister repair outcomes, thereby supporting the original conclusions of Adamo et al. and Bickel et al. that BRC-1 and SMC-5/6 function in homolog-independent repair. Importantly, we observed intersister repair in *brc-1* and *smc-5* mutants arising from both SPO-11 independent (ICR assay) and SPO-11 dependent (EdU labeling assay) DSBs, indicating that intersister repair in these mutants is not artifactual to Mos1 excision-derived DSBs but instead is engaged to resolve both endogenous and exogenous DNA damage.

A growing body of evidence has found that BRC-1 and SMC-5/6 exert subtle but important functions regulating interhomolog repair. BRC-1 physically interacts with the synaptonemal complex, localizes to recombination nodules, promotes efficient interhomolog crossover formation, and regulates the meiotic crossover landscape under conditions of meiotic dysfunction (*Li et al., 2018*; *Janisiw et al., 2018*; *Li et al., 2020*). Our observation that interhomolog recombinants in *brc-1* were elevated in the IH assay specifically at the transition between the interhomolog and non-interhomolog windows parallels recent cytological analysis (*Trivedi et al., 2022*) indicating that BRC-1 promotes timely meiotic progression, perhaps by regulating efficient interhomolog crossover recombination. We identified a similar increase in IH assay recombinants at the end of the interhomolog window in *smc-5* mutants, raising the possibility that SMC-5/6 may also function to promote meiotic progression.

SMC-5/6 acts redundantly with the BLM helicase HIM-6 and the RMI1 homolog RMH-1 to prevent or eliminate interchromosomal (including interhomolog) attachments (*Hong et al., 2016*; *Jagut et al., 2016*). In *Arabidopsis*, loss-of-function mutations in multiple SMC5/6 subunits reduces crossover

interference; similarly, *smc-5;him-6 C. elegans* mutants form additional interhomolog crossovers with reduced interference (*Hong et al., 2016*; *Zhu et al., 2021*). Taking our data in the context of this accumulating evidence, we suggest that BRC-1 and SMC-5/6 have diverse functions for regulating multiple pathways of DSB repair, including intersister recombination, interhomolog recombination, and the engagement of error-prone pathways.

## Functions of BRC-1 in *C. elegans* meiotic DNA repair

We find that meiotic cells deficient in BRC-1 exhibit multiple DNA repair defects, including reduced noncrossover conversion tract length, elevated rates of intersister crossovers, and engagement of error prone DSB repair mechanisms at the mid-pachytene stage. What functions of BRC-1 may underpin these phenotypes? BRCA1 is thought to regulate many early steps in recombination including DSB resection, strand invasion, and D-loop formation in other model systems (*Kamp et al., 2020*; *Zhao et al., 2017*; *Chen et al., 2008*; *Cruz-García et al., 2014*; *Chandramouly et al., 2013*). We propose that many of these functions are conserved in *C. elegans* BRC-1.

While research in budding yeast, mammalian systems, and *Arabidopsis* suggests that SDSA is the primary pathway for the formation of noncrossovers in meiosis (*Hunter, 2015*; *Ahuja et al., 2021*; *Marsolier-Kergoat et al., 2018*) and that processing of joint molecular intermediates can generate noncrossovers during *Drosophila* meiosis (*Crown et al., 2014*), the mechanisms by which *C. elegans* noncrossover recombination occurs is unknown. Our finding that *brc-1* mutation affects the extent of ICR assay noncrossover gene conversion, but not crossover gene conversion, suggests that homolog-independent noncrossovers arise from a distinct intermediate or undergo differential processing from crossovers in *C. elegans*. This result is consistent with a model in which either SDSA or joint molecule dissolution is a primary mechanism of intersister noncrossover recombination in the *C. elegans* germline (*Figure 4—figure supplement 6*).

The size of an SDSA or dissolution noncrossover conversion tract is primarily determined by the length of DNA strand extension (*Marsolier-Kergoat et al., 2018*; *Guo et al., 2017*; *Keelagher et al., 2011*). Human BRCA1 promotes strand invasion and D-loop formation (*Zhao et al., 2017*), which may influence the efficiency of strand extension. Our conversion tract data therefore raises the possibility that *C. elegans* BRC-1 influences the formation and/or stability of strand invasion intermediates, thereby promoting the formation of long ICR assay noncrossover gene conversion events (*Figure 4—figure supplement 6*). Our data also demonstrate that *brc-1* mutants exhibit elevated intersister crossovers. If BRC-1 only functions to promote strand invasion and D-loop formation, then we would expect *brc-1* mutation to reduce intersister crossovers and not increase their occurrence. Previous studies have also suggested that BRCA1/BRC-1 regulates DSB resection (*Chen et al., 2008*; *Cruz-García et al., 2014*), and we propose that this function better accounts for our observed increase in intersister crossovers in the *brc-1* null mutant. Specifically, BRCA1-promoted long range DSB resection may be important for the efficiency of SDSA by ensuring sufficient single stranded DNA is exposed on the second end of the DSB to facilitate strand annealing (*Kamp et al., 2020*; *Chandramouly et al., 2013*). While sufficient resection may be critical in resolving SDSA noncrossovers, long range resection is not required for the efficient formation of joint molecules (*Zakharyevich et al., 2010*). Thus, reduced length of DNA resection due to a *brc-1* mutation may impede SDSA and therefore increase the probability that DSBs will form joint molecule intermediates, thereby promoting intersister crossover outcomes.

Reduced resection in conjunction with inefficient strand invasion and synthesis during recombination may further explain the ectopic engagement of TMEJ observed in *brc-1* mutants (*Kamp et al., 2020*). Short range resection provides sufficient substrate for TMEJ (*Ramsden et al., 2022*), which in combination with inefficient homology search may provide more opportunity for TMEJ engagement. Polymerase θ helicase activity is also sufficient to remove recombinases from single-stranded DNA to promote end-joining repair (*Schaub et al., 2022*). BRC-1 may promote recombination and prevent TMEJ by stabilizing RAD-51 on resected DNA or antagonizing this activity of polymerase θ. Notably, the timepoint at which we identified mutants in the *brc-1* ICR and IH assays correlate to the stage of meiotic prophase I at which BRC-1 is also required in late meiotic prophase I for the loading and/or maintenance of RAD-51 (*Li et al., 2018*; *Janisiw et al., 2018*). Overall, our data is consistent with a model in which BRC-1 promotes multiple DSB repair steps, including resection and the formation of early strand invasion intermediates, to facilitate intersister noncrossover repair (*Figure 4—figure supplement 6*).

## Functions of SMC-5/6 in *C. elegans* meiotic DSB repair

*C. elegans* deficient in SMC-5/6 exhibit severe meiotic genome instability and become sterile over successive generations (*Volkova et al., 2020*; *Bickel et al., 2010*). Despite these stark phenotypes, our ICR and IH assays identified only relatively subtle phenotypes in *smc-5* mutants. It should be noted, however, that the ICR and IH assays only detect DSB repair outcomes that encode a functional protein product. Thus, many of the severe DSB repair mutations associated with SMC-5/6 deficiency (*Volkova et al., 2020*) may disrupt the coding sequence in the ICR or IH assays and therefore escape detection.

In budding yeast, Smc5/6 prevents the accumulation of toxic interchromosomal attachments and recombination intermediates (*Bonner et al., 2016*; *Chen et al., 2009*; *Peng et al., 2018*; *Xaver et al., 2013*; *Lilienthal et al., 2013*; *Copsey et al., 2013*). Prior evidence in *C. elegans* suggests that some of these functions are likely conserved, as double mutants for *smc-5* and the BLM helicase homolog *him-6* are sterile and display chromatin bridges indicative of persistent interchromosomal attachments (*Hong et al., 2016*). This synthetic phenotype suggests that these two complexes act in parallel to prevent the accumulation of joint molecules. A previous study (*Almanzar et al., 2021*) and the data we present here reveal that both SMC-5 and HIM-6 repress intersister crossovers. The synthetic sterility associated with loss of both SMC-5 and HIM-6 then may be a product of parallel functions for these proteins in limiting and/or resolving joint molecules. Although BLM is known to play multiple roles in regulating recombination, a core function of this helicase is in antagonism of joint molecule formation and promotion of noncrossover recombination (*Schvarzstein et al., 2014*; *McVey et al., 2004*; *Weinert and Rio, 2007*). SMC-5/6 in *C. elegans* meiosis may therefore act as a second line of defense to ensure the elimination of inappropriate joint molecule intermediates that have formed more stable configurations. We also did not observe a large increase in ICR assay crossover products in *smc-6* mutants (*Figure 2—figure supplement 1B*). Thus, it is possible that even in the absence of fully assembled SMC-5/6 complex, SMC-5 and SMC-6 may independently exert some functions that subtly impact DSB repair outcomes.

Our observation that *smc-5* mutation does not alter ICR assay conversion tracts is also consistent with a model in which SMC-5/6 influences recombination following joint molecule formation. Recent work has shown that SMC5/6 is capable of DNA loop-extrusion, indicating a function by which the complex may organize chromatin to facilitate efficient DSB repair (*Pradhan et al., 2022*). Specific subunits of SMC-5/6 also exhibit enzymatic function, such as the E3 SUMO ligase Nse2/Mms21 (*Andrews et al., 2005*), suggesting that SMC-5/6 may act to post-translationally modify target proteins to regulate DNA repair. In summary, our data indicates that SMC-5/6 is not necessary for *C. elegans* intersister recombination and opens the door to future investigation into its molecular roles in regulating efficient DSB repair.

## Temporal regulation of error-prone meiotic DSB repair

In the ICR and IH assays that we performed in *brc-1* mutants, we identified mutagenic repair events specifically at the 22–34 hr timepoint, corresponding to oocytes in mid pachytene at the time of Mos1-excision induced DSB formation. Further, the repair events that we identified frequently displayed microhomologies flanking the deletion site – a characteristic signature of TMEJ. The limited temporal window in which we identified these events suggests that TMEJ may be relegated to later stages of meiotic prophase I. There are a number of important events that coincide with the mid/late pachytene transition of *C. elegans* meiosis, including a MAP kinase phosphorylation cascade, designation of interhomolog crossovers, a switch from RAD-50 dependence to independence for loading of RAD-51 to resected DNA, and loss of access to the homolog as a ready repair template (*Rosu et al., 2011*; *Yokoo et al., 2012*; *Hayashi et al., 2007*; *Nadarajan et al., 2016*; *Church et al., 1995*; *Lee et al., 2007*; *Kritikou et al., 2006*). In mitotic human cell culture, Polθ is phosphorylated and activated in specific stages of mitosis by polo-like kinase 1 (PLK1; *Gelot et al., 2023*). The *C. elegans* meiotic polo-like kinase ortholog PLK-2 acts in mid-late pachytene to antagonize CHK-2 activity, thereby promoting interhomolog crossover designation and halting DSB induction (*Zhang et al., 2023*). We raise the hypothesis that *C. elegans* meiotic POLQ-1 may be activated in late stages of meiotic prophase I by PLK-2 analogously to mitotic human Polθ activation by PLK1. These regulatory events may confer a switch in cellular 'priorities' from ensuring formation of interhomolog recombination intermediates to promoting repair of all residual DSBs even through error prone mechanisms. By repairing all residual

DSBs (even in the wake of sequence errors), germ cells avoid catastrophic chromosome fragmentation during the meiotic divisions.

During the mid to late pachytene transition, an important function of BRC-1 (and to a lesser extent SMC-5/6) may be to prevent TMEJ either by antagonizing this pathway or facilitating efficient recombination. Our irradiation experiments revealed that both *brc-1* and *smc-5* mutant oocytes exhibit greater sensitivity to exogenous DNA damage in late stages of prophase I, suggesting that cellular requirements for efficacious DSB repair change during the transition to late pachytene. Moreover, during the mid-late pachytene stage, several changes regarding BRC-1 occur: (1) BRC-1 protein localization changes; and, (2) BRC-1 is required to load (and/or stabilize) RAD-51 filaments (*Li et al., 2018*; *Janisiw et al., 2018*; *Figure 4—figure supplement 2C*). We found that *brc-1* mutants incur mutations with characteristic TMEJ signatures specifically at the mid/late pachytene stage, suggesting that the changes in BRC-1 localization and function at this stage may coincide with changes in the availability and/or regulation of error prone repair mechanisms. Our irradiation experiments demonstrated that *smc-5;brc-1* double mutant oocytes in late meiotic prophase I are dependent upon TMEJ DNA polymerase θ homolog *polq-1* for viability. If BRC-1 functions that repress TMEJ (*Kamp et al., 2020*) are specific to late prophase, then this result suggests that many DSBs in *smc-5;brc-1* mutants induced in early prophase may not be repaired until mid/late pachytene, when TMEJ is active. Spatiotemporal transcriptomic analysis has shown that *polq-1* is expressed throughout meiotic prophase I (*Tzur et al., 2018*). As we only identified error-prone resolution of DSBs induced at mid pachytene, our findings raise the possibility that BRC-1 independent mechanisms may repress TMEJ in early/mid pachytene. Our results in *brc-1* mutants therefore lay the groundwork for future research delineating the temporal regulation of error-prone meiotic DSB repair. Taken together, our study reveals that the engagement of error-prone and recombination DSB repair pathways are differentially regulated during the course of *C. elegans* meiotic prophase I.

## Interaction between BRC-1 and SMC-5/6 in meiotic DNA repair

Our observation of multiple genetic interactions in *brc-1* and *smc-5* double mutants, but no coincident change in either SMC-5/6 or BRC-1 localization in their respective null mutants, suggests that the phenotypes of *smc-5*, *brc-1*, and *smc-5;brc-1* mutants are likely not derived from direct physical interactions between these complexes nor action on shared substrates. Instead, we propose that BRC-1 and SMC-5/6 exert sequential roles in regulation of DSB repair. A similar model was proposed by *Hong et al., 2016* which postulated that early recombination defects in *brc-1* mutants may alleviate the toxic recombination intermediates formed in *smc-5;him-6* double mutants.

How might DNA repair defects in *brc-1* mutants ameliorate genomic instability associated with *smc-5* mutation? Our observation that *brc-1* mutation suppresses the increased RAD-51 foci in *smc-5* mutants suggests that BRC-1 acts to promote recombination repair of DSBs even in contexts where recombination is defective (such as in *smc-5* null mutants). Ablation of NHEJ and TMEJ proteins in *brc-1* mutants rescues RAD-51 foci formation, indicating that BRC-1 acts to promote recombinase loading at the exclusion of error-prone DSB repair (*Trivedi et al., 2022*). Thus, it is likely that many DSBs in *smc-5;brc-1* mutants are repaired by recombination-independent pathways and circumvent *smc-5* mutant defects.

*brc-1* mutant defects in DSB resection and strand invasion may also contribute to suppressing *smc-5* mutant phenotypes. In budding yeast, the additional ssDNA generated by long range resectioning of a DSB is used for homology search (*Chung et al., 2010*). Inefficient resection in *brc-1* mutants may reduce the extent of homology that could anneal to heterologous templates and contribute to toxic joint molecules (*Figure 4—figure supplement 6*). Compromised strand invasion and D-loop formation in *brc-1* mutants could also limit the capacity for DSBs to form multi-chromatid engagements. Conversely, resection defects of *brc-1* mutants may increase the risk for toxic recombination intermediates in *smc-5* mutants by limiting the efficiency of SDSA and therefore biasing DSBs to form joint molecules. Future studies assessing the specific recombination intermediates formed in *brc-1* and *smc-5* mutants is needed to distinguish between these possibilities.

In summary, our data suggest that BRC-1 acts upstream of SMC-5/6 to promote recombination by facilitating early DSB processing and strand invasion. SMC-5/6, then, likely functions downstream of BRC-1 to stabilize or regulate joint molecules, consistent with its functions in preventing the accumulation of toxic interchromosomal attachments (*Hong et al., 2016*; *Jagut et al., 2016*). Our work

reveals interplay between these highly conserved DNA repair complexes in regulating DNA repair pathway engagement and outlines targets for future research to determine the molecular basis of these interactions.

## Materials and methods
### *Caenorhabditis elegans* strains and maintenance

*Caenorhabditis elegans* strains were maintained at 15°C or 20°C on nematode growth medium (NGM) plates seeded with OP50 bacteria. All experiments were performed in the N2 genetic background of *C. elegans* and animals were maintained at 20 °C for at least two generations preceding an experiment.
 Strains used in this study include:

N2 (wild type)
AV554 (*dpy-13(e184sd) unc-5(ox171::Mos1)*/ nT1 (*qIs51*) *IV*; krIs14 (*phsp-16.48::MosTransposase; lin-15B; punc-122::GFP*) / nT1 (*qIs51*) *V*)
CB791 (*unc-5(e791) IV*),
DLW14 (*unc-5(lib1*[ICR assay *pmyo-3::GFP(-); unc-119(+); pmyo-2::GFP(Mos1)]*) *IV*; krIs14 (*phsp-16.48::MosTransposase; lin-15B; punc-122::GFP*) *V*)
DLW23 (*smc-5(ok2421)*/mIn1 [*dpy-10(e128) mIs14*] *II*; *unc-5(lib1*[ICR assay *pmyo-3::GFP(-); unc-119(+); pmyo-2::GFP(Mos1)]*) *IV*; krIs14 (*phsp-16.48::MosTransposase; lin-15B; punc-122::GFP*) *V*)
DLW76 (*brc-1(tm1145) brd-1(dw1) III*; *unc-5(lib1*[ICR assay *pmyo-3::GFP(-); unc-119(+); pmyo-2::GFP(Mos1)]*) *IV*; krIs14 (*phsp-16.48::MosTransposase; lin-15B; punc-122::GFP*) *V*)DLW81 (*smc-5(ok2421)*/mIn1[*dpy-10(e128) mIs14*] *II*; *unc-5(e791) IV*)
DLW100 (*brc-1(tm1145) brd-1(dw1) III*; *unc-5(e791) IV*)DLW131 (*smc-5(ok2421)*/mIn1[*dpy-10(e128) mIs14*] *II*; *lig-4(ok716) brc-1(xoe4) III*)
DLW134 (*smc-5(ok2421)*/mIn1[*dpy-10(e128) mIs14*] *II*; *polq-1(tm2572) brc-1(xoe4) III*)
DLW137 (*smc-5(ok2421)*/mIn1 [*mIs14 dpy-10(e128)*] *II*; *brc-1(xoe4) III*)
DLW157 (*brc-1(xoe4) III*; *unc-5(e791) IV*)
DLW175 (*smc-5(syb3065* [::AID*::3xFLAG]) *II*; *brc-1(xoe4) III*)
DLW182 (*smc-5(ok2421)*/mIn1[*dpy-10(e128) mIs14*] *II*; *GFP::brc-1 III*)
DLW202 (*smc-5(ok2421)*/mIn1 [*dpy-10(e128) mIs14*] *II*; *dpy-13(e184sd) unc-5(ox171::Mos1) IV*; krIs14 [*phsp-16.48::MosTransposase; lin-15B?; punc-122::GFP*] *V*)
DLW203 (*brc-1(xoe4) III*; *dpy-13(e184sd) unc-5(ox171::Mos1) IV*; krIs14 [*phsp-16.48::MosTransposase; lin-15B; punc-122::GFP*] *V*)
DLW220 (*smc-5(syb3065[::AID*::3xFLAG]) ieSi65 [sun-1p::TIR1::sun-1 3'UTR, Cbr-unc-119(+)] II*)
DLW221 (*msh-2(ok2410)*/tmC20[*unc-14(tmIs1219) dpy-5(tm9715)*] *I*)
DLW222 (*smc-6(ok3294)*/mIn1 [*dpy-10(e128) mIs14*] *II*; *unc-5(lib1*[ICR assay *pmyo-3::GFP(-); unc-119(+); pmyo-2::GFP(Mos1)]*) *IV*; krIs14 (*phsp-16.48::MosTransposase; lin-15B; punc-122::GFP*) *V*)
DLW223 (*smc-6(ok3294)*/mIn1[*dpy-10(e128) mIs14*] *II*; *unc-5(e791) IV*)
DLW231 (*msh-2(ok2410)*/tmC20[*unc-14(tmIs1219) dpy-5(tm9715)*] *I*; *unc-5(lib1*[ICR assay *pmyo-3::GFP(-); unc-119(+); pmyo-2::GFP(Mos1)]*) *IV*; krIs14 (*phsp-16.48::MosTransposase; lin-15B; punc-122::GFP*) *V*)
DLW232 (*msh-2(ok2410)*/tmC20[*unc-14(tmIs1219) dpy-5(tm9715)*] *I*; *unc-5(e791) IV*)
DLW243 (*smc-5(syb3065[::AID*::3xFLAG]) ieSi65 [sun-1p::TIR1::sun-1 3'UTR, Cbr-unc-119(+)] II*; *brc-1(xoe4) III*; *unc-5(lib1*[ICR assay *pmyo-3::GFP(-); unc-119(+); pmyo-2::GFP(Mos1)]*) *IV*; krIs14 (*phsp-16.48::MosTransposase; lin-15B; punc-122::GFP*) *V*)
DLW244 (*smc-5(syb3065[::AID*::3xFLAG]) ieSi65 [sun-1p::TIR1::sun-1 3'UTR, Cbr-unc-119(+)] II*; *brc-1(xoe4) III*; *unc-5(e791) IV*)
DLW247 (*smc-5(syb3065[::AID*::3xFLAG]) ieSi65 [sun-1p::TIR1::sun-1 3'UTR, Cbr-unc-119(+)] II*; *brc-1(xoe4) III*)
DW102 (*brc-1(tm1145) brd-1(dw1) III*)
JEL515 (*GFP::brc-1 III*)
JEL730 (*brc-1(xoe4) III*)
PHX3065 (*smc-5(syb3065* [::AID*::3xFLAG]) *II*)

YE57 (*smc-5(ok2421)*)/mln1 [mls14 *dpy-10(e128)*] *II*)
YE58 (*smc-6(ok3294)*)/mln1 [mls14 *dpy-10(e128)*] *II*)

Double and triple mutants that carried the *smc-5(ok2421)* and *brc-1(xoe4)* alleles incurred mutations within ~6–10 generations of propagation, as indicated by progeny with movement defects, body morphology defects, or the presence of male offspring. To minimize the risk of de novo suppressor or enhancer mutations influencing the phenotypes we observed in these mutants, we froze stocks of these double and triple mutants at –80 °C within three generations of the strains' construction. All experiments using these strains were carried out on stocks which had been maintained for less than 1–2 months. If a strain began to segregate mutant phenotypes, a new isolate of the freshly generated strain was thawed from frozen stocks.

## CRISPR/Cas9 genome editing

CRISPR/Cas9 genome editing was performed by SUNY Biotech to generate the *smc-5(syb3065)* allele in which the endogenous sequence of *smc-5* is modified at its C terminus to code for both an AID* tag (peptide sequence PKDPAKPPAKAQVVGWPPVRSYRKNVMVSCKSSGGPEAAAFVK) and a 3xFLAG tag (peptide sequence DYKDHDGDYKDHDIDYKDDDDK). The coding sequence of *smc-5*, the AID* tag, and the 3xFLAG tag were respectively connected by flexible GAGS peptide linkers. The repair template for this insertion was synthesized as a single strand oligo and was injected with Cas9 enzyme and a single guide RNA targeting the 12th exon of the *smc-5* locus. Successful integration was confirmed by PCR and Sanger sequencing. CRISPR edited strains were backcrossed three times to N2 before experiments were performed.

## *C. elegans* brood viability assays and Bayesian hierarchical modeling analysis

L4 stage hermaphrodite nematodes of each genotype to be scored were isolated 16–18 hr before irradiation was to be performed and were maintained at 15 °C on NGM plates seeded with OP50. These worms were then exposed to 0, 2500, or 5000 Rads of ionizing radiation from a $Cs^{137}$ source (University of Oregon). Following irradiation, n=5 hermaphrodites of each genotype and treatment combination were placed onto individual NGM plates seeded with OP50 and were maintained at 20 °C. If plates were to be mated, n=3 young adult N2 males were also added to the plate at this timepoint and were discarded at the 10 hr transfer. At 10 hr, 22 hr, and 46 hr post irradiation, the irradiated hermaphrodites were transferred to new NGM plates seeded with OP50. Fifty-eight hours after irradiation, the parent hermaphrodites were discarded. The proportion of F1 progeny that hatched, did not hatch ('dead eggs' indicating embryonic lethality), or were unfertilized on each plate was scored 36–48 hr after the removal of the parent hermaphrodite from a plate. The brood size of each hermaphrodite was calculated as (hatched progeny) + (dead eggs). Brood viability at each timepoint was calculated as (hatched progeny) / (brood size). Brood viability experiments were conducted in triplicate for the following genotypes: unmated N2, unmated *smc-5*, unmated *brc-1*, unmated *smc-5;brc-1*, unmated *smc-5;brc-1;lig-4*, and unmated *smc-5;brc-1;polq-1*. Brood viability assays were conducted in duplicate for the following genotypes: mated *smc-5;brc-1;polq-1*, mated *polq-1*, and unmated *polq-1* and unmated *smc-5(syb3065)*. Brood viability was assessed in one replicate for mated *smc-5;brc-1* mutants.

Brood viabilities of individual hermaphrodites for each given genotype and irradiation treatment were analyzed using RStan (**Stan Development Team, 2021**). The brood viability data of individual hermaphrodites (h) for each genotype (g), timepoint scored (t), and irradiation treatment (i) was fit to a Beta-Binomial model. To facilitate model fitting, the Beta distribution was parameterized using the mean (φ) and shape parameter sum ($\lambda$).

$$\phi_{g,t,i} = \frac{\alpha_{g,t,i}}{(\alpha_{g,t,i} + \beta_{g,t,i})}$$

$$\lambda_{g,t,i} = \alpha_{g,t,i} + \beta_{g,t,i}$$

$$p_{g,t,i} \sim \text{Beta}(\lambda_{g,t,i} \times \phi_{g,t,i}, \lambda_{g,t,i} \times (1 - \phi_{g,t,i}))$$

$$\text{Hatched Progeny}_{g,t,i,h} \sim \text{Binomial}(n = \text{Brood size}_{g,t,i,h}, p_{g,t,i})$$

A metric (termed "gamma") for the effect of ionizing radiation on the observed brood viability of each genotype was calculated in the Generated Quantities block during MCMC sampling from the posterior probability distribution of the parameter p, defined as:

$$\text{gamma}_{g,t,i} = \frac{p_{g,t,i}}{p_{g,t,0\,\text{Rads}}}$$

In addition to the model fit statistics output from Stan, model fit was assessed by posterior simulations. The expected brood viability for 1500 parent hermaphrodites from each genotype, timepoint, and irradiation treatment were simulated (*Figure 4—figure supplement 3C*). For each simulated parent hermaphrodite, a brood size was simulated from a Poisson distribution with a rate parameter equal to the mean brood size of the corresponding experimental group, values for φ and $\lambda$ were sampled from the respective posterior probability distributions, and a value for p was simulated from a Beta distribution with of the sampled φ and $\lambda$. The number of hatched progeny were simulated ~Binomial (brood size, p).

## Intersister/intrachromatid repair assay (ICR Assay)

ICR assays were performed as described in *Toraason et al., 2021c*; *Toraason et al., 2021b*. Parent (P0) hermaphrodites for the ICR assay for each genotype were generated by crossing (see cross schemes detailed below).

ICR assay cross schemes:

1. Wild type (N2): P0 hermaphrodites were generated by crossing: (1) N2 males to DLW14 hermaphrodites to generate *unc-5(lib1)/+IV; krIs14/+V* males; (2) F1 males to CB791 hermaphrodites to generate *unc-5(lib1)/unc-5(e791) IV; krIs14/+V* hermaphrodites.
2. *brc-1* mutant: P0 hermaphrodites were generated by crossing: (1) JEL730 males to DLW156 hermaphrodites to generate *brc-1(xoe4) III; unc-5(lib1)/+IV; krIs14/+V* males; (2) F1 males to DLW157 hermaphrodites to generate *brc-1(xoe4) III; unc-5(lib1)/unc-5(e791) IV; krIs14/+V* hermaphrodites.
3. *smc-5* mutant: P0 hermaphrodites were generated by crossing: (1) YE57 males to DLW23 hermaphrodites to generate *smc-5(ok2421)/mIn1 II; unc-5(lib1)/+IV; krIs14/+V* males; (2) F1 males to DLW81 hermaphrodites to generate *smc-5(ok2421) II; unc-5(lib1)/unc-5(e791) IV; krIs14/+V* hermaphrodites.
4. *brc-1;brd-1* mutant: P0 hermaphrodites were generated by crossing: (1) DW102 males to DLW76 hermaphrodites to generate *brc-1(tm1145) brd-1(dw1) III;; unc-5(lib1)/+IV; krIs14/+V* males; (2) F1 males to DLW100 hermaphrodites to generate *brc-1(tm1145) brd-1(dw1) III; unc-5(lib1)/unc-5(e791) IV; krIs14/+V* hermaphrodites.
5. *smc-6* mutant: P0 hermaphrodites were generated by crossing: (1) YE58 males to DLW222 hermaphrodites to generate *smc-6(ok3294)/mIn1 II; unc-5(lib1)/+IV; krIs14/+V* males; (2) F1 males to DLW223 hermaphrodites to generate *smc-6(ok3294) II; unc-5(lib1)/unc-5(e791) IV; krIs14/+V* hermaphrodites.
6. *msh-2* mutant: P0 hermaphrodites were generated by crossing: (1) YE57 males to DLW23 hermaphrodites to generate *msh-2(ok2410)/tmC20 I; unc-5(lib1)/+IV; krIs14/+V* males; (2) F1 males to DLW81 hermaphrodites to generate *msh-2(ok2410); unc-5(lib1)/unc-5(e791) IV; krIs14/+V* hermaphrodites.
7. *smc-5::AID\*::3xFLAG;brc-1* mutant: P0 hermaphrodites were generated by crossing: (1) DLW220 males to DLW243 hermaphrodites to generate *smc-5(syb3065) ieSi51 II; brc-1(xoe4) III; unc-5(lib1)/+IV; krIs14/+V* males; (2) F1 males to DLW244 hermaphrodites to generate *smc-5(syb3065) ieSi51 II; brc-1(xoe4) III; unc-5(lib1)/unc-5(e791) IV; krIs14/+V* hermaphrodites.

In brief, P0 hermaphrodites of the desired genotype were isolated 16–18 hr before heat shock and were maintained at 15°C. For this step and all subsequent transfers, worms were maintained on NGM plates seeded with OP50 with the exception of *smc-5(syb3065);brc-1(xoe4)* mutants, which were transferred to NGM plates with 10 mM auxin 16–18 hr before heat shock and were maintained on plates containing 10 mM auxin at all subsequent steps. Heat shock was performed in an air incubator (refrigerated Peltier incubator, VWR Model VR16P) for one hour. The P0 worms were then allowed to recover at 20°C for nine hours. For wildtype, *brc-1(xoe4)*, and *smc-5(ok2421)* ICR assays, P0 hermaphrodites were placed onto individual plates and were maintained at 20°C and were transferred to new plates at 22 hr, 34 hr, and 46 hr after heat shock. Fifty-eight hours after heat shock, these P0 hermaphrodites

were removed from their NGM plates and discarded. *smc-6(ok3294)*, *brc-1(tm1145);brd-1(dw1)*, and *msh-2(ok2410)* assays were performed identically, except that n=3 P0 hermaphrodites were placed onto individual plates and P0 hermaphrodites were transferred to new plates 22 hr after heat shock and were discarded 46 hr after heat shock. For *smc-5(syb3065);brc-1(xoe4)* assays, n=3 P0 hermaphrodites were placed onto individual plates and were transferred 22 hr, 34 hr, and 46 hr after heat shock and then were discarded 58 hr after heat shock. In all assays, plates with P0 hermaphrodites were maintained at 20°C, while plates with F1 progeny were placed at 15 °C.

F1 progeny were scored for GFP fluorescence ~54–70 hr after the P0 hermaphrodite was removed. Approximately 18 hr before scoring, plates with F1 progeny were placed at 25°C to enhance GFP expression. Fluorescent phenotype scoring was performed on a Axio Zoom v16 fluorescence stereoscope (Zeiss). F1 progeny which expressed recombinant fluorescence phenotypes were isolated and lysed for sequencing (see Sequencing and analysis of ICR assay conversion tracts). Nonrecombinant progeny were discarded. If all progeny on a plate were in larval developmental stages at the time of scoring, then the number of dead eggs and unfertilized oocytes were additionally recorded.

Due to the strict temporal timeline of the ICR assay (*Toraason et al., 2021b*; *Toraason et al., 2021c*, see above), these experiments were not performed in parallel. All displayed frequencies represent the combined sums of multiple replicates. ICR assays in *brc-1(xoe4)*, *smc-5(ok2421)*, and *smc-5(syb3065);brc-1* mutants were replicated 4 times, *brc-1(tm1145);brd-1(dw1)* and *smc-6(ok3294)* mutants were replicated 3 times, and *msh-2(ok2410)* mutants were replicated twice. In all assays, the broods of at least 20 parent hermaphrodites were scored in each replicate. The ICR assay in a wild type genetic background was performed once and combined with previous data (*Toraason et al., 2021c*). For detailed information on the number of progeny scored in individual replicates, see *Figure 1—source data 1*, *Figure 2—source data 1*, and *Figure 4—source data 1*.

## Sequencing and analysis of ICR assay conversion tracts

Recombinant ICR assay progeny were placed in 10 µL of 1 x Worm Lysis Buffer for lysis (50 mM KCl, 100 mM TricHCl pH 8.2, 2.5 mM MgCl$_2$, 0.45% IGEPAL, 0.45% Tween20, 0.3 µg/µL proteinase K in ddH$_2$O) and were iteratively frozen and thawed three times in a dry ice and 95% EtOH bath and a 65 °C water bath. Samples were then incubated at 60°C for 1 hr and 95°C for 15 min to inactive the proteinase K. Final lysates were diluted with 10 µL ddH2O.

Conversion tracts were PCR amplified using OneTaq 2 x Master Mix (New England Biolabs). Noncrossover recombination products were amplified using forward primer DLO822 (5'-ATTTTAAC CCTTCGGGGTACG-3') and reverse primer DLO823 (5'-TCCATGCCATGTGTTAATCCCA-3'). Crossover recombination products were amplified using forward primer DLO824 (5'-AGATCCATCTAG AAATGCCGGT-3') and reverse primer DLO546 (5'-AGTTGGTAATGGTAGCGACC-3'). PCR products were run on an Agarose gel and desired bands were isolated by gel extraction (QIAquick Gel Extraction Kit, New England Biolabs) and were eluted in ddH$_2$O. Amplicons were submitted for Sanger sequencing (Sequetech) with three primers. Noncrossovers were sequenced using DLO822, DLO823, and DLO1077 (5'-CACGGAACAGGTAGGTTTTCCA-3') and crossovers were sequenced using DLO824, DLO546, and DLO1077.

Sanger sequencing chromatograms were analyzed using Benchling alignment software (Benchling) to determine converted polymorphisms. Heteroduplex DNA signals were identified by two prominent peaks in the chromatogram at the site of a known polymorphism. Putative heteroduplex samples were PCR amplified and submitted for sequencing a second time for confirmation as described above.

Samples that produced PCR products of the expected size but did not yield interpretable sequencing were subsequently analyzed using TOPO cloned amplicons. ICR assay locus amplicons were PCR amplified as described above but were immediately cloned into pCR2.1 vector using the Original TOPO-TA Cloning Kit (Invitrogen) following kit instructions. Putative successful amplicon clones were identified by PCR amplification using 2xOneTaq Master Mix (New England Biolabs) with primers DLO883 (5'-CAGGAAACAGCTATGACCATG-3') and DLO884 (5'-TGTTAAAACGACGGCC AGGT-3'). Plasmids containing amplicon inserts were isolated from 2 mL LB + Amp cultures using the GENEJET Miniprep kit (Thermo Fischer Scientific) and were submitted for Sanger sequencing (Sequetech) using primers DLO883 and DLO884.

To acquire additional wild type ICR assay crossover tracts for our analyses, three 'bulk' replicates of the wild type ICR assay were performed following the protocol described in the 'Intersister/

intrachromatid repair assay' with the following exceptions: (1) n=3 hermaphrodites were passaged together on individual plates during the experiment; (2) transfers were only performed at 10 hr, 22 hr, and 46 hr following heat shock; and, (3) plates were screened for body wall GFP+ crossover recombinants but the frequency of pharynx GFP+ and GFP- nonrecombinant progeny were not scored. Body wall GFP+ crossover progeny were lysed following the preceding protocol.

Not all lysed recombinant yielded successful PCR products or sequences. Of the additional wild type ICR assay recombinants sequenced for this manuscript, 11 of 11 noncrossover and 52 of 52 crossover lysates were successfully sequenced. Among lysates from *brc-1* mutant ICR assays, 37 of 37 noncrossover and 70 of 73 crossover lysates were successfully sequenced. Among lysates from *smc-5* mutant ICR assays, 56 of 56 noncrossover and 27 of 28 crossover lysates were successfully sequenced.

## Interhomolog assay (IH assay)

IH assays were performed as described in *Rosu et al., 2011*. In brief, P0 hermaphrodites were generated by crossing (see cross schemes detailed below). The IH assay was performed twice for each genotype.

IH assay cross schemes:

1. Wild type (N2): P0 hermaphrodites were generated by crossing: (1) N2 males to AV554 hermaphrodites to generate *dpy-13(e184sd) unc-5(ox171::Mos1)*/+IV; krIs14/+V males; (2) F1 males to CB791 hermaphrodites to generate *dpy-13(e184sd) unc-5(ox171::Mos1)*/*unc-5(e791)* IV; krIs14/+V hermaphrodites.
2. *brc-1* mutant: P0 hermaphrodites were generated by crossing: (1) JEL730 males to DLW203 hermaphrodites to generate *brc-1(xoe4)* III; *dpy-13(e184sd) unc-5(ox171::Mos1)*/+IV; krIs14/+V males; (2) F1 males to DLW157 hermaphrodites to generate *brc-1(xoe4)* III; *dpy-13(e184sd) unc-5(ox171::Mos1)*/*unc-5(e791)* IV; krIs14/+V hermaphrodites.
3. *smc-5* mutant: P0 hermaphrodites were generated by crossing: (1) YE57 males to DLW202 hermaphrodites to generate *smc-5(ok2421)*/mIn1 II; *dpy-13(e184sd) unc-5(ox171::Mos1)*/+IV; krIs14/+V males; (2) F1 males to DLW81 hermaphrodites to generate *smc-5(ok2421)* II; *dpy-13(e184sd) unc-5(ox171::Mos1)*/*unc-5(e791)* IV; krIs14/+V hermaphrodites.

The heat shock and timing at which parent hermaphrodites were transferred to new NGM plates was performed identically to the ICR assay (see 'Intersister/Intrachromatid repair assay (ICR assay)' above). However, the number of eggs and unfertilized oocytes laid by each hermaphrodite was recorded immediately following the removal of the parent hermaphrodite at each timepoint and plates carrying F1 progeny were maintained at 20 °C. Plates were scored for F1 wild type moving (non-Unc) progeny ~84–96 hr after parent hermaphrodites were removed. F1 Unc progeny were discarded.

F1 non-Unc progeny were placed on single NGM plates seeded with OP50 bacteria. Dpy non-Unc progeny (putative noncrossover recombinants) were lysed following the protocol described in 'Sequencing and analysis of SCR assay conversion tracts'. If Dpy non-Unc progeny died before the time of lysis and had laid F2 progeny, non-Unc segregant F2s were lysed instead. Non-Dpy non-Unc progeny (potential crossover recombinants) were allowed to lay F2 progeny. If progeny were laid and Dpy non-Unc F2 segregants were identified, these Dpy non-Unc F2s were lysed and the F1 was inferred not to be a crossover recombinant. If >50 F2 progeny were on the plate and no Dpy non-Unc segregants were identified, the F1 was assumed to be a crossover recombinant and no worms were lysed. If very few progeny were laid and no Dpy non-Unc segregants were identified, the F1 non-Unc or its non-Unc F2 offspring were lysed and subsequently subjected to PCR genotyping analysis using OneTaq 2 x Master Mix (New England Biolabs) to determine the genotype of *unc-5* and *dpy-13*. The presence of Mos1 in the *unc-5* locus was assessed using primers DLO987 (5'-TCTTCTTGCCAAAGCG ATTC-3') and DLO1082 (5'-TTCTCTCCGAGCAATGTTCC-3'). The *dpy-13* locus was assessed using primers DLO151 (5'-ATTCCGGATGCGAGGGAT-3') and DLO152 (5'-TCTCCTCGCAAGGCTTCTGT -3'). Lysed F1 nUnc nDpy progeny were inferred to be crossover recombinants if the worms (1) carried the Mos1 transposon at the *unc-5* locus and were heterozygous for the *dpy-13(e184)* allele, or; (2) did not carry the Mos1 transposon at the *unc-5* locus and were homozygous wild type for *dpy-13*.

The *unc-5* locus was amplified for sequencing by PCR using OneTaq 2 x Master Mix (NEB) with primers DLO1081 (5'-TCTTTTCAGGCTTTGGCACTG-3') and DLO1082. PCR products were run on an agarose gel and desired bands were isolated by gel extraction (QIAquick Gel Extraction Kit, New England Biolabs) and were eluted in ddH$_2$O. These amplicons were submitted for Sanger sequencing

(Sequetech) with primer DLO1082 or DLO150 (5'-GTTCCATGTTTGATGCTCCAAAAG-3'). Sanger sequencing chromatograms were compared to the wild type *unc-5* sequence using Benchling alignment software. Samples that showed a reversion to wild-type *unc-5* sequence at the site of Mos1 excision were inferred to be noncrossover recombinants. Samples that showed mutations that preserved the reading frame of the *unc-5* locus were considered 'mutant non-Unc'. One of the five *brc-1* IH assay mutant non-Uncs we sequenced carried two distinguishable mutagenic repair products. These two mutations likely represent the outcomes of both a meiotic DSB repair event and an additional somatic repair event in the progeny. We have previously observed analogous somatic Mos1 excision events in F1 progeny in the ICR assay (*Toraason et al., 2021c*; *Toraason et al., 2021b*). As we cannot distinguish the source of the respective repair events, this mutant was excluded from subsequent sequence analysis.

Samples which showed mixed sequences despite a clear amplicon being generated in the PCR were subsequently TOPO cloned, as described in 'Sequencing and analysis of ICR assay conversion tracts', except that the amplicon used in the reaction was generated using primers DLO1081 and DLO1082.

Not all interhomolog assay non-Unc progeny were able to be confirmed as recombinants by sequencing. Of the wild type IH assay non-Unc progeny identified, 176 of 178 putative noncrossovers were successfully sequenced. Among lysates from *brc-1* mutant IH assays, 72 of 76 putative noncrossovers were successfully sequenced. Among lysates from *smc-5* mutant IH assays, 213 of 229 putative noncrossovers were successfully sequenced. Non-Unc progeny whose *unc-5* DNA repair events could not be identified by sequencing were considered 'undetermined non-Unc' in subsequent analyses of this data.

## EdU sister chromatid exchange assay

EdU Sister Chromatid Exchange assays were performed as described in *Almanzar et al., 2021*; *Almanzar et al., 2022*. For the *smc-5* null mutants, uninterpretable bivalents were excluded due to stochasticity in EdU incorporation, progression of nuclei in the gonad, and orientation of the bivalent in the xy-axis of the microscope.

## Immunofluorescence localization of SMC-5/6 and BRC-1

Immunofluorescence was performed as in *Libuda et al., 2013* or *Howe et al., 2001*. For both protocols, L4 staged hermaphrodites were isolated 18–22 hours before dissection and maintained at 20 °C on NGM plates seeded with OP50. Nematodes that were irradiated preceding an immunofluorescence experiment were exposed to a $Cs^{137}$ source (University of Oregon) and were dissected less than an hour after irradiation. Samples prepared for GFP::BRC-1 or RAD-51 visualization were dissected in 1 x Egg Buffer (118mMNaCl, 48mM $KCl_2$, 2mM $CaCl_2$, 2mM $MgCl_2$, 25mM HEPES pH7.4, 0.1% Tween20) and were fixed in 1 x Egg Buffer with 1% paraformaldehyde for 5 min on a Superfrost Plus slide (VWR). Samples prepared for SMC-5::AID*::3xFLAG visualization were dissected in 1 x Sperm Salts (50mM PIPES pH7, 25mM KCl, 1mM $MgSO_4$, 45mM NaCl, 2mM $CaCl_2$) and an equal volume of 1 x Sperm Salts with 3% paraformaldehyde was applied for 5 min before samples were affixed to a Superfrost Plus slide (VWR). For both protocols, gonads were then flash frozen in liquid nitrogen and the cover slip was removed. Germlines stained for GFP::BRC-1 or RAD-51 were then fixed for 1 min in ice cold MeOH and then were washed in 1 x PBST (1 x PBS, 0.1% Tween20), while germlines stained for SMC-5::AID*::3xFLAG were fixed for 1 min in ice cold 95% EtOH and then were washed in 1xPBST* (1 x PBS, 0.5% Triton-X100, 1 mM EDTA pH8). Slides were then washed 3 x in PBST or PBST* respectively before being placed in Block (1xPBST or 1xPBST* with 0.7% bovine serum albumin) for at least 1 hr.

Primary antibody staining was performed by placing 50 µL of antibody diluted in PBST for samples in which GFP::BRC-1 or RAD-51 were to be visualized or PBST* if the sample was to be stained for SMC-5::AID*::3xFLAG (see below for specific dilutions of primary antibodies). A parafilm coverslip was placed on each sample and the slides were incubated for 16–18 hr in a dark humidifying chamber. Slides were then washed 3 x in PBST or PBST* for 10 min. Fifty µL of secondary antibody diluted in PBST for samples in which GFP::BRC-1 or RAD-51 were to be visualized or PBST* if the sample was to be stained for SMC-5::AID*::3xFLAG (see below for specific dilutions of primary antibodies) was then placed on each slide. Slides were incubated for 2 hr in a dark humidifying chamber with a parafilm

coverslip. Slides were then washed 3 x in PBST or PBST* for 10 min in a dark chamber. 50µL of 2µg/mL DAPI was then applied to each slide. Slides were incubated in a dark humidifying chamber with parafilm coverslips for 5 min. Slides were then washed 1 x in PBST or PBST* for 5 min in a dark chamber before being mounted in VectaShield with a No. 1.5 coverslip (VWR) and sealed with nail polish.

Slides were maintained at 4°C until imaging. All slides stained for SMC-5::AID*::3xFLAG were imaged within 48 hr of mounting. Immunofluorescence images were acquired at 512x512 pixel dimensions on an Applied Precision DeltaVision microscope. All images were acquired in 3D using Z-stacks at 0.2µm intervals and were deconvolved with Applied Precision softWoRx deconvolution software. Individual images of whole germlines were stitched as 3D Z-stacks in FIJI using the Grid/Collection Stitching plugin (*Preibisch et al., 2009*) or as maximum intensity projections using Photoshop (Adobe). The intensity levels of images displayed in this manuscript were adjusted in Photoshop for clarity.

The following primary antibodies were used in this study at the given dilutions: Chicken αRAD-51 (1:1000; *Kurhanewicz et al., 2020*), Rabbit αRAD-51 (1:5000, *Kurhanewicz et al., 2020*), Mouse αmini-AID M214-3 (1:500, MBL International), Rat αHTP-3 (1:1000, this study), Rabbit αGFP (1:500 *Yokoo et al., 2012*). The following secondary antibodies were used in this study at the given dilutions: Alexa Fluor 488 Goat αRabbit (1:200, Invitrogen A21428), Alexa Fluor 488 Goat αMouse (1:200, Invitrogen A11001), Alexa Fluor 555 Goat αRat (1:200, Invitrogen A48263) Alexa Fluor 647 Goat αChicken (1:200, Invitrogen A21449).

Immunofluorescence staining of SMC-5/6 was further attempted using previously published antibodies (*Bickel et al., 2010*). We were unable to generate samples with specific staining using these antibodies, potentially due to their age. We additionally attempted to raise new antibodies using the previously published epitopes (*Bickel et al., 2010*) (synthesized by GenScript) in chickens (SMC-5) or rabbits (SMC-6). Neither of these antibodies exhibited specific staining.

## RAD-51 foci quantification

RAD- 51 foci per nucleus were quantified using Imaris (Bitplane) as described in detail in *Toraason et al., 2021a*. RAD-51 foci were defined as Spot objects (estimated XY diameter 0.1, model PSF-elongation 1.37, background subtraction enabled) and nuclei were defined using the DAPI channel as Surface objects (smooth 0.1–0.15, background 3–4, seed point diameter 2.5–4). Manual thresholding was performed to best represent the foci and nuclei in each image. The number of RAD-51 foci per nucleus was calculated using the 'losest Distance to Surface' statistic to find the total number of Spots within 0.4µm of a given nucleus. The length of each germline was delineated using the Measurement Points tool, and specific points were placed at the beginning of the transition zone, end of the transition zone, and end of pachytene. Nuclei positions were transformed from 3D coordinates to a liner order as previously described (*Toraason et al., 2021a*). The length of each germline was normalized to pachytene such that the beginning of pachytene was position 0 and the end of pachytene was position 1.

## Antibody generation

The HTP-3 antibody used in this study was generated from an identical C-terminal segment of the HTP-3 protein (synthesized by GenScript) as was used by *MacQueen et al., 2005*. Antibodies were produced in rats and affinity purified by Pocono Rabbit Farms.

## Statistics

All statistics were calculated in R (v4.0.3). Specific tests utilized are described in text or in the figure legends. Data wrangling was performed using the Tidyverse (v1.3.0) package (*Wickham et al., 2019*). Bayesian hierarchical models were fit using Rstan (v2.21.2) (*Stan Development Team, 2021*). Binomial Confidence Intervals were calculated using the DescTools package (v 0.99.38).

## Acknowledgements

We thank the CGC (funded by National Institutes of Health P40 OD010440) for providing strains. We also thank J Engebrecht for generously sharing strains carrying the *brc-1(xoe4)* and *GFP::brc-1* alleles. We thank C Cahoon, A Naftaly, and N Kurhanewicz for thoughtful comments on this manuscript. We also thank G Csankovszki for sharing antibodies for SMC-5 and SMC-6. This work was supported by the National Institutes of Health T32GM007413 to ET and JEB; Advancing Science in America (ARCS)

Foundation Award to ET; National Institutes of Health R25HD070817 to AS; Genetics Training Grant T32M007464 to DEA; a Pilot Project Award from the American Cancer Society, R35GM128804 grant from NIGMS, and start-up funds from the University of Utah to OR; and National Institutes of Health R00HD076165 and R35GM128890 to DEL. DEL is also a recipient of a March of Dimes Basil O'Connor Starter Scholar award and Searle Scholar Award.

## Additional information

### Funding

| Funder | Grant reference number | Author |
|---|---|---|
| National Institute of General Medical Sciences | R35GM128890 | Diana E Libuda |
| Eunice Kennedy Shriver National Institute of Child Health and Human Development | R00HD076165 | Diana E Libuda |
| National Institute of General Medical Sciences | T32GM007413 | Erik Toraason |
| Eunice Kennedy Shriver National Institute of Child Health and Human Development | R25HD070817 | Alina Salagean |
| National Institute of General Medical Sciences | T32GM007464 | David E Almanzar |
| National Institute of General Medical Sciences | R35GM128804 | Ofer Rog |
| Advancing Science in America | Foundation Award | Erik Toraason |
| American Cancer Society | Pilot Project Award | Ofer Rog |
| University of Utah | start-up funds | Ofer Rog |

The funders had no role in study design, data collection and interpretation, or the decision to submit the work for publication.

### Author contributions

Erik Toraason, Conceptualization, Formal analysis, Investigation, Visualization, Methodology, Writing – original draft, Writing – review and editing; Alina Salagean, David E Almanzar, Investigation, Writing – review and editing; Jordan E Brown, Colette M Richter, Nicole A Kurhanewicz, Investigation; Ofer Rog, Supervision, Funding acquisition, Writing – review and editing; Diana E Libuda, Conceptualization, Formal analysis, Supervision, Funding acquisition, Writing – original draft, Project administration, Writing – review and editing

### Author ORCIDs

Erik Toraason http://orcid.org/0000-0003-0500-955X
Alina Salagean https://orcid.org/0000-0002-5377-9375
Nicole A Kurhanewicz http://orcid.org/0000-0003-2023-9310
Ofer Rog https://orcid.org/0000-0001-6558-6194
Diana E Libuda https://orcid.org/0000-0002-4944-1814

### Decision letter and Author response

Decision letter https://doi.org/10.7554/eLife.80687.sa1
Author response https://doi.org/10.7554/eLife.80687.sa2

## Additional files

### Supplementary files
• MDAR checklist

### Data availability
All data generated or analyzed in this study are included in the manuscript and supporting files. Source data files have been provided for Figures 1A, 1C, 1D, 1E, 2A, 2B, 2C, 2D, 3, 4, Figure 1 - figure supplements 2-3, Figure 2 - figure supplements 1-2, and Figure 4 - figure supplements 1-3. Source code files have been provided for Figure 4 - figure supplement 3.

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

# Appendix 1

## Appendix 1—key resources table

| Reagent type (species) or resource | Designation | Source or reference | Identifiers | Additional information |
|---|---|---|---|---|
| Gene (*C. elegans*) | *brc-1* | https://wormbase.org/species/c_elegans/gene/WBGene00000264 | WormBase ID: WBGene00000264 | |
| Gene (*C. elegans*) | *brd-1* | https://wormbase.org/species/c_elegans/gene/WBGene00000265 | WormBase ID: WBGene00000265 | |
| Gene (*C. elegans*) | *smc-5* | https://wormbase.org/species/c_elegans/gene/WBGene00016151 | WormBase ID: WBGene00016151 | |
| Gene (*C. elegans*) | *smc-6* | https://wormbase.org/species/c_elegans/gene/WBGene00010056 | WormBase ID: WBGene00010056 | |
| Gene (*C. elegans*) | *polq-1* | https://wormbase.org/species/c_elegans/gene/WBGene00020964 | WormBase ID: WBGene00020964 | |
| Gene (*C. elegans*) | *lig-4* | https://wormbase.org/species/c_elegans/gene/WBGene00002986 | WormBase ID: WBGene00002986 | |
| Gene (*C. elegans*) | *unc-5* | https://wormbase.org/species/c_elegans/gene/WBGene00006745 | WormBase ID: WBGene00006745 | |
| Gene (*C. elegans*) | *rad-51* | https://wormbase.org/species/c_elegans/gene/WBGene00004297 | WormBase ID: WBGene00004297 | |
| Strain, strain background (*C. elegans*) | For *C. elegans* alleles and strain information, see strain list in Methods (*C. elegans* strains and maintenance) | This paper | | See strain list in Methods (*C. elegans* strains and maintenance) |
| Genetic reagent (*C. elegans*) | Details on *C. elegans* CRISPR/Cas9 editing are detailed in the Methods (CRISPR/Cas9 genome editing) | This paper | | CRISPR/Cas9 transgenics were generated by InVivo Biosystems |
| Antibody | Anti-RAD-51 (chicken polyclonal) | *Kurhanewicz et al., 2020* | | IF (1:1000) |
| Antibody | Anti-RAD-51 (rabbit polyclonal) | *Yokoo et al., 2012* | | IF (1:5000) |
| Antibody | Anti-mini-AID M214-3 (mouse monoclonal) | MBL International | | IF (1:500) |
| Antibody | Anti-HTP-3 (rat polyclonal) | This study | | IF (1:1000) |
| Antibody | Anti-GFP (rabbit polyclonal) | *Yokoo et al., 2012* | | IF (1:500) |
| Antibody | Alexa Fluor 488 anti-rabbit (goat polyclonal) | Thermo Fisher | Cat# A21428 | IF (1:200) |
| Antibody | Alexa Fluor 488 anti-mouse (goat polyclonal) | Thermo Fisher | Cat# A11001 | IF (1:200) |
| Antibody | Alexa Fluor 555 anti-rat (goat polyclonal) | Thermo Fisher | Cat# A48263 | IF (1:200) |
| Antibody | Alexa Fluor 647 anti-chicken (goat polyclonal) | Thermo Fisher | Cat# A21449 | IF (1:200) |
| Sequence-based reagent | PCR primers for amplifying ICR and IH assay recombinants are described in Methods (Sequencing and analysis of ICR assay conversion tracts, Interhomolog assay (IH assay)) | *Toraason et al., 2021c*, This Paper | PCR Primers | See Methods (Sequencing and analysis of ICR assay conversion tracts, Interhomolog assay (IH assay)) |
| Chemical compound, drug | Naphthaleneacetic acid (auxin) | PhytoTechnology Laboratories | Cat# N610 | 10 mM |
| Software, algorithm | Whole Gonad Analysis (R script) v1.0 | *Toraason et al., 2021a*; *Toraason et al., 2021d* | https://github.com/libudalab/Gonad-Analysis-Pipeline | |
| Software, algorithm | RStan | Stan Development Team | https://mc-stan.org/ | |
| Software, algorithm | Imaris 9 | Oxford Instruments | https://imaris.oxinst.com/products | |
| Software, algorithm | FIJI | *Schindelin et al., 2012* | https://imagej.net/software/fiji/ | |
| Software, algorithm | FIJI plug in – Stitcher | *Preibisch et al., 2009* | https://imagej.net/plugins/image-stitching | |

