## [Editor Report]

This study provides an important insight in the regulation of DSB repair during meiosis in *C. elegans*. The analysis is based on genetic, genomic and cytological approaches allowing to monitoring distinct DNA repair pathways and providing compelling evidence for their regulation and genetic controls.

---

## [Decision Letter]

**Decision letter after peer review:**

Thank you for submitting your article "BRCA1/BRC-1 and SMC-5/6 regulate DNA repair pathway engagement during *C. elegans* meiosis" for consideration by *eLife*. Your article has been reviewed by 3 peer reviewers, including Bernard de Massy as the Reviewing Editor and Reviewer #1, and the evaluation has been overseen by Jessica Tyler as the Senior Editor. The following individual involved in the review of your submission has agreed to reveal their identity: Anton Gartner (Reviewer #2).

The reviewers have discussed their reviews with one another, and the Reviewing Editor has drafted this synthetic review with "Essential revisions" and "Other revisions" to help you prepare a revised submission.

Essential revisions:

1) Understanding the assays is important for the readers. Even though the information is available in previous publications, a supplementary figure describing the assays (ICR and IH) and recombination products and their detection is suggested.

2) As noted by the authors, the inter-sister CO increase was not expected and seems to contradict the previous publications by Adamo et al. and Bickel et al. Have the authors thought about testing intersister recombination with another allele of their mutants to exclude effects due to unknown mutations? This issue needs to be addressed and discussed.

3) L303-304: The interpretation of the increased GFP progeny is a major issue and cannot be redirected to a few sentences in the methods section as this is the major topic of this study. The methods section outlines several alternatives, whereas the main text seems to prioritize some. This seems arbitrary and should be clarified with experimental data (for instance testing the SNP hypothesis).

4) The authors cannot conclude that both assays (L 323 support a function for SMC-5 in repressing intersister crossover in *C. elegans* meiosis: Data from Edu is supported by only 6 bivalents coming from 2 oocytes. In addition, could the authors demonstrate that those exchanges are taking place during meiosis? (as performed for him-6, this may require to analyze a smc-5 spo-11 mutant).

5) L361: the potential delay should be shown. Assigning the two time-windows as inter homolog and non interhomolog is at the basis of the experimental approach and whether it is altered in the mutants or not must be known and taken into account for interpretations.

6) Analysis of ICR and IH in brc-1 smc-5 double mutant. The authors could use balancers for both loci to maintain the strain, and minimize accumulation of genomic instability (from the strain list, a balancer was used for smc-5 but not for brc-1). Alternatively, the Smc-5 degron allele could be used.

7) Irradiation experiments: Please provide cytological data of DSB repair detection (ie RPA or Rad-51) in the different mutant contexts and information about the number of DSBs induced at the different doses of IR (14 DSB per homolog pair for 5000 rads as from Almanzar et al?). Also provide information about the γ sensitivity of germ cells of wt, single and double mutants.

The readout of the irradiation experiments is brood viability. One major problem with the interpretation is that sperm is also irradiated and can contribute to the phenotype. To avoid this issue, the authors should irradiate hermaphrodites and mate with non-irradiated males. This could be tested at least for one genotype (ie brc-1 smc-5) to evaluate the potential impact of sperm irradiation.

8) The main figure (figure 4) should present brood viability, and the fit to the model as supplementary figure, as qualitatively the effects can be visualized by examining the brood viability: i) greater sensitivity of smc-5 compared to brc-1 (the claim for a greater sensitivity in the non-interhomolog window is not convincing as the differences of the γ parameter are quite small, may be excepted for the brc-1 mutant at 5000 rads; in addition, no statistical test is validating the differences),ii) dispensability of BRC-1 at early prophase in low IR, iii) suppression of smc-5 sensitivity by brc-1 in the interhomolog window, iv) sensitization by polq-1.

9) Discussion:

The discussion is mostly reiterating the data from the results and should be more focused on the conclusions that can be drawn, novel points or questions to be addressed, and relating to current knowledge on Brc-1 and Smc-5 in other species.

The question of chromosome fragmentation in Smc-5 and discussion about the data from this manuscript (L 674…) is unclear. How could the mutations detected in smc-5 (Volkova et al.) explain the efficient recovery of HR events by the ICR and IH assays? As indicated above (point 2), the authors should discuss the previous study concluding that Brc-1 is required for intersister recombination (Adamo et al., 2008).

After citing this work in the introduction, the authors ignore this opposite conclusion from the discussion. May be the chromatid fragments in brc-1 msh-4 are products of TMEJ ? Similarly, Bickel et al. 2010, proposed that Smc-5 is required for homolog-independent (presumably intersister) events. The authors must solve this apparent contradiction.

The question of potential differences between Spo11 and Mos1 induced breaks should be discussed and thus with alternative interpretations.

10) The model presented in Figure 5, with the molecular intermediates is only indirectly supported by the data which does not allow determining which steps of the reaction are affected besides for the resection phenotype from the gene conversion tract analysis. The model would be better suited to be presented as a supplementary figure.

11) Box plots: Plot data points instead of box plots: Figure 1A, 1c, supfig2b, 2d, Figure 2B, 2d, Figure 3C, d

Other revisions:

12) Abstract: L21: SMC-5 was not shown to limit error prone repair.

13) The authors should introduce more widely the current knowledge about Brca1 and Smc5/6 with information from other species. Their results should also be integrated into a larger context.

14) It is unclear whether each mutant was analyzed in parallel with a wild-type control. In principle to control for experimental variations (temperature…) this should be done. As presented in figures, one has the impression that wild-type and mutants were done in parallel. Please explain clearly the experimental procedures in the methods, and provide relevant information in the legends.

15) In several figures (ie Figure 1C and others), sample sizes and duplicates are not clearly indicated: how many animals, how many oocytes. It would be easier to have N values on the figures. It is mentioned in the methods that ICR assays were repeated 4 times with at least 20 parents each (what about the IH assays ?), but the source data shows only the total pool of progeny tested. What is the inter-parent and inter-replicate variation?

16) For counting bivalents, some with visually interpretable bivalents were counted. Likely some bivalents either DAPI or Edu staining was difficult to interpret. How to exclude a selection bias in this analysis?

17) Fig1D: some numbers do not match the table: 1/41 instead of 1/43…, same for 1E.

And the mutant non-unc is 17.2% in the figure but 13.2% in the text.

18) Figure 3 A and C: The number of noncrossover conversion tracts in the non-IH window for smc-5 and brc-1 seems to be too low (with one long event in brc-1) to be statistically valid, even though a p value was calculated by the Fisher's exact test. The authors should dampen their conclusion that Brc-1 only affects tract length in the IH window or alternatively increase sample size.

19)

L 39: SPo11 references: Keeney et al. 1997; Bergerat et al. 1997.

L 44: The use of "resolve" for DSB is odd (also L 55, 74).

L 51: "In SDSA, one resected end" might be better than "the resected end".

L 74: Utilize homologous recombination.

L 104-5: vague; not additive could mean synergistic or unchanged.

L 121 (and L514-515). For clarity, it would be good to use 'more genetic language. Eg 'the hypersensitivity of smc-5 mutations towards IR is suppressed/bypassed by brc-1 (mutation). L514: Could the term resilience be avoided, replaced by Eg:.…strains or more sensitive to IR than WT,.. are hypersensitive to IR,.. the IR sensitivity of smc-5 mutants is suppressed by brc-1….

L 176-177: seem out of place, please revise.

L179: "Does not necessarily represent recombinant progeny". The meaning is unclear. Please clarify.

Line 198 and elsewhere: "is not required for" could be understood to mean "is not involved in." The intended meaning is "is not essential for."

L 210. More frequently is vague please provide quantitative information.

L 246: "a slight delay in the rate of meiotic progression" the rate is not delayed.

L 261: "at the time of Mos1 excision" perhaps should say "at the time of heat shock" since there is likely some delay between heat shock and transposase expression and transposon excision.

L 269 suppl figure 3B. It would be better to precisely refer to the events shown in the text. Eg label (1-4) and refer to (3B.1) 3B.2 etc etc.

L 283: 5 not 4 mutants have been identified (Figure 1E), but only 4 are reported?

Line 299: "SMC-5/6 is therefore not required for viability in *C. elegans*…" this conclusion does not follow from the data presented in the preceding sentences.

L 318: intersister in smc-5. "Among a limited sample", please clarify, does it refer to 6 individuals?

Supplemental Figure 3B: using a monospaced font in the DNA sequences would ensure that the nucleotides in each strand were properly aligned.

Figure 2C-D: it would be helpful to see an image of an "uninterpretable" bivalent in the SCE assay and to know what percentage of bivalents were discarded as uninterpretable in wt, brc-1, and smc-5 animals.

Line 380: could be more specific. "in restricting/limiting intersister CO recombination".

L 389: Figure 3A not 4A.

L 428: This conclusion is confusing. Brca-1 mutant were shown above to have more CO (intersister). How can one conclude that it does not regulate CO recombination?

L 448: Where is the heteroduplex DNA information? Please provide this data.

Figure 3C and D (graphs): font is too small.

Figure 4, supplemental figure 4: it would be interesting to see controls with for example polq-1 single mutants and polq-1 double mutants with brc-1 and smc-5. These controls might shed some light on what is going on and get closer to mechanism.

Sup Figure 4: What is the sample size ? Wild type: no point is visible; brc-1, only 1 point is visible at 0 rad ? Please clarify. In panel B, the simulation is probably the distribution shown in black, please clarify.

In some cases (smc-5 2500 rad) the simulation does not fit the data. This is problematic because the validity of the γ parameter is therefore uncertain. The source data do not show the individual data points (ie brood viability of individual hermaphrodites), please revise.

Supplemental Figure 7 is never referred to in the manuscript.

What is smc-5 (phx3065) ? PHX3065 seems to be a strain name.

L 642: "influences noncrossover intermediate processing" is vague.

Are the most important strains for measuring inter-sister and interhomolog recombination deposited in the GCG strain collection?

*Reviewer #1 (Recommendations for the authors):*

Figure 1: The number of biological duplicates is unclear and single-point data cannot be found in the tables. It is said that assays were replicated 4 times and at least 20 parents each, but sup table 1 only show the total progeny. What is the inter-parent and inter-replicate variation?

Fig1D: some numbers do not match the table: 1/41 instead of 1/43.…, same for 1E

And the mutant non-unc is 17.2% in the figure but 13.2% in the text ?

L171: "Does not necessarily represent recombinant progeny". The meaning is unclear. Please clarify.

L 222. More frequently is vague please provide quantitative information.

L 252: 5 not 4 mutants have been identified, but only 4 are reported?

Interhomolog assay: the progeny is referred to as non-unc or unc-5 (+). This is confusing , please use one of the two. It seems that referring to unc-5 (+) is better even if this can include some mutational events.

L228 and sup2A: increase of interhomolog !!

L287: The interpretation of the increased GFP progeny is a major issue and cannot be redirected to a few sentences in the methods section as this is the major topic of this study. The methods section outlines several alternatives, whereas the main text seems to prioritize some. This seems arbitrary and should be clarified with experimental data (for instance testing the SNP hypothesis).

L324: intersister in smc-5. "Among a limited sample", please clarify, does it refer to 6 individuals?

L 232: What are the other defects?

L338: the potential delay should be shown.

L343: It is not because the mutant has no defect in the IH homolog assay that smc-5 does not have a function in IH repair. Please revise.

Figure 2: A and B: why 2 graphs: the data 10-22hr is plotted twice.

L365: Figure 3A not 4A

Figure 3C and D (graphs) : letter sizes are too small

L388: The numbers for non-crossovers in the IH window is too low (8 events) to draw a convincing comparison between wt and brc-1.

L402: This conclusion is confusing. Brca-1 mutant were shown above to have more CO (intersister). How can one conclude that it does not regulate CO recombination?

L422: Where is the heteroduplex DNA information? Please provide.

Sup Figure 4: What is the sample size ? Wild type: no point is visible; brc-1, only 1 point is visible at 0 rad ? Please clarify. In panel B, the simulation is probably the distribution shown in black, please clarify. In some cases (smc-5 2500 rad) the simulation does not fit the data. This is problematic because the validity of the γ parameter is therefore uncertain. The sup. table 3 and 4 do not show the individual data points, please revise.

What is smc-5 (phx3065) ? PHX3065 seems to be a strain name.

Figure 4: A DSB repair defective control would be important to evaluate how severe the defect in smc-5 mutant is, may be Rad-51 or Rad-54 mutants could be used. The conclusion that the sensitivity (estimated by Γ parameter, 4A) is greater in the non-interhomolog window is not supported by a statistical test. Please revise.

The authors should analyze the phenotypes of the brc-1 smc-5 double mutant with the ICR and IH assays.

Also a more direct monitoring of DSB repair must be done, for instance by detection of RPA and/or RAD51 foci in the different mutants (especially double and triple mutants) with and without the different doses of radiation. How many breaks are induced by 2500 and 5000 rads ?

L 40: SPo11 references: Keeney et al. 1997; Bergerat et al. 1997

L 45: The use of "resolve" for DSB is odd (also L 58)

L75: utilize homologous recombination

Comment on the Discussion

The discussion is mostly reiterating the data from the results and should be more focused on the conclusions that can be drawn, novel points or questions to be addressed.

The question of chromosome fragmentation in Smc-5 and discussion about the data from this manuscript (L 674…) is unclear. How could the mutations detected in smc-5 (Volkova et al.) explain the efficient recovery of HR events by the ICR and IH assays?

The authors should discuss the previous study concluding that Brc-1 is required for intersister recombination (Adamo et al., 2008). After citing this work in the introduction, the authors ignore this opposite conclusion from the discussion. May be the chromatid fragments in brc-1 msh-4 are products of TMEJ ?

Similarly, Bickel et al. 2010, proposed that Smc-5 is required for homolog-independent (presumably intersister) events. The authors must solve this apparent contradiction. Differences between assays and/or Spo11 vs Mos1 induced breaks or other features should be tested.

The model presented in Figure 5 is not supported by the data which does not allow determining which steps of the reaction are affected beside a support for a resection phenotype from the gene conversion tract analysis. The authors could speculate on the different alternatives for the activities of Brc-1 and Smc-5 (and they are many) but presenting a synthetic model is misleading not coherent with the content of the data. It should be removed. If the authors want to keep this visualization, it should be in supplementary data.

A schematic representation of the key assays: the meiotic progression in the gonad to show the reverse kinetic principle, the ICR and the IH assays as a sup figure would help understanding the data.

*Reviewer #2 (Recommendations for the authors):*

Importantly, data on sister chromatid repair by CO and non-CO data are both supported by GFP reporter assays as well as by directly observing sister chromatid exchanges of EDU-marked chromatids. Having this said, the reviewer struggled to deduce how the reporter system used, allows for distinguishing CO and non-CO outcomes. Could a Suppl model; Figure showing the detailed intermediates be provided? I did not find this in the original *C. elegans* paper either.

Would it be possible to analyze sister chromatid recombination (proportion of CO versus non-CO outcome as measured by reporter assay) also in brc-1 smc-5 double mutants?

A weakness in the study is that key phenotypes are only analyzed by one allele of brc1 and smc-5 each. Would it be possible to repeat a key experiment, eg EDU labelling or GFP reporter with a second allele? This way, the possibility of secondary, possibly unlinked mutations contributing to phenotypes can be formally excluded. Nowadays, a simple genome edit (generating a second allele) might do the trick.

In the radiation sensitivity assays, would it be possible to also include an analysis of sensitivities of single and double mutants of mitotically proliferating germ cells (before the enter prophase). Such analysis would very much add to our understanding of differential pathway usage of germ cells.

Figures 1-3, I think it would be good, wherever possible, in graphs and diagrams (Figure 1a, c, Figure 2a, b, Figure 3 c, d) to show individual data points in addition to error bars.

Line (46) Is the statement that the majority of meiotic DSBs are repaired by interhomolog recombination true, also for *C. elegans*?

Lines 121. I think, for clarity, it would be good to use 'more genetic language. Eg 'the hypersensitivity of smc-5 mutations towards IR is suppressed/bypassed by brc-1 (mutation).

Generally, whenever possible only describe mutant phenotypes, or protein function but not both within a sentence.

Line 269 suppl figure 3B. It would be good to precisely refer to the events shown in the text. Eg label (1-5) and refer to (3B.1) 3B.2 etc.

Line 403, Figure 3C and D. Be uniform in what you label as short and long conversion tracks.

Lines 514, and 515, same as the above comment, line 121.

Page 25 consider rewording. Can the word 'resilience' be avoided?

Eg:.…strains or more sensitive to IR than WT,.. are hypersensitive to IR,.. the IR sensitivity of smc-5 mutants is suppressed by brc-1…

Line 587. 599 can this section be reworded to avoid the word 'normal'? Eg BRC-1 localization did not change upon IR in…..

Are the most important strains for measuring inter-sister and interhomolog recombination deposited in the GCG strain collection?

*Reviewer #3 (Recommendations for the authors):*

As currently written, the manuscript is very focused on *C. elegans* meiosis. The experiments are generally well done, and the results will be of interest to investigators studying DNA repair and meiotic recombination in *C. elegans*. However, it is unclear whether the results are unique to *C. elegans*, and the authors do not give sufficient context to convincingly frame their conclusions as novel or identifying conserved meiotic mechanisms.

At least in the merged PDF of the manuscript, lines 176 and 177 seem out of place, and reading from line 147 (before Figure 1) to line 176 (after Figure 1) does not make sense. I think lines 176-77 should be deleted, but do they belong somewhere else? Also, the first reference to Figure 1A comes later in the text than I would have expected, so I'm not sure if other text was missing that should have been there.

The ICR and IH assays are so important to this paper that there should be a supplemental figure showing how they work and give different outcomes for CO vs NCO, and where the polymorphisms are. Even though they have been published elsewhere, it would be helpful not to have to find those papers to be able to follow the experiments done here.

Line 461: How do the doses of IR used affect the number of DSBs or RAD-51 foci? Are these doses expected to increase breaks 2-fold? 10-fold? The experiments are difficult to interpret without knowing how DSB number (or some DSB proxy) changes.

As the authors acknowledge, there is substantial variation in viability between the broods of different irradiated (and even unirradiated) hermaphrodites with the same genotype. The cause of this variability is not known. However, it is possible that sperm defects contribute to the variability. Irradiation of young adult hermaphrodites will induce mutations both in the germline nuclei undergoing meiosis and in the several hundred mature sperm made earlier in development. Thus, the "reverse timecourse" following irradiation is capturing both the effects of mutations made at different stages of nuclei undergoing oogenic meiosis and sperm that had already completed meiosis when irradiated. Larval lethality could result from either, and may be revealing how an embryo copes with inheriting damaged DNA from the sperm. It is also formally possible that sperm differences account for the variability observed in unirradiated worms. It is already difficult to compare the data using Mos1 induced breaks to the irradiation data because the assays are completely different. However, the Mos data also presumably is only measuring recombination during oocyte production (i.e. not in sperm), while larval lethality following irradiation is likely measuring damage in sperm as well as during oogenesis. I would recommend mating irradiated hermaphrodites with unirradiated males to simplify the analysis by looking only at consequences of damage that will ultimately affect the oocyte.

The irradiation/larval lethality assay was used instead of the ICR/IH assays because "genetically balanced smc-5;brc-1 double mutants can still acquire mutations and become progressively sterile over the course of a few generations" (line 456). From the strain list, it looks like only smc-5 was balanced, but brc-1 was maintained in a homozygous condition. Given the availability of numerous small inversion balancers made by the Mitani lab, it should be possible to balance both loci. This may prevent the accumulation of defects and allow the ICR/IH assays to be done in double mutants. Alternatively, perhaps the degron-tagged smc-5 allele could be used to deplete SMC-5 acutely. If so, this would greatly strengthen the paper.

For me, the way the irradiation data are presented calls into question their robustness, or at least suggests that the conclusions are only as good as the mathematical model. I think many of the key results are visible in Supplemental Figure 4A, at least at some doses. As a biology person and not a statistics person, I would find it substantially more convincing to discuss the empirical data first, then bring in the model data to support the empirical data. At the very least, I need better justification for the design of the statistical model and its reliability.

Identical data are presented as wild-type controls in several figures (Figures1A and 4A, 1B and 4C, Supplemental Figure 1A and B, and elsewhere). This gives the false impression that each mutant was analyzed in parallel with a wild-type control. However, it is unclear from the text, figure legends, and methods whether any of the experiments were done with parallel controls. In fact, the wild-type dataset was performed once and combined with previous data (line 950). Ideally, wild-type and mutants would be analyzed at the same time, on the same batch of plates, same batch of worm food, and in the same incubator, since meiosis is temperature sensitive, and heat shock genes can be induced by a variety of stressors, including wet plates. If this was not done, the authors should clearly state it in the methods and ideally also in the figure legends, since it is an important point. I also think it would be less confusing, and would potentially make the paper easier to read, if some figures were combined, for example 1A and 4A could be one figure (with a legend stating that multiple replicates were done, but not always in parallel for different genotypes); 1B and 4C could easily be combined.

Figures 1B and 4C: the N values are given for number of bivalents analyzed. The number for WT seems reasonable, but the numbers for mutants seem small even though this is a somewhat difficult experiment. How many bivalents were analyzed in each oocyte, and how were they chosen? How many oocytes were analyzed? How many animals?

Figure 3A and C: it is hard to interpret the data showing a statistically significant effect of brc-1 depletion on resection tract length because so many fewer NCO tracts were analyzed in brc-1 than in wt or smc-5 and only one long tract was identified. I guess this would only mean that the magnitude of the effect is being underestimated, not the opposite.

---

## [Author Response]

Essential revisions:1) Understanding the assays is important for the readers. Even though the information is available in previous publications, a supplementary figure describing the assays (ICR and IH) and recombination products and their detection is suggested.

We appreciate the reviewers’ helpful suggestion and have generated a new supplemental figure (Figure 1 —figure supplement 1) with diagrams of the IH and ICR assays.

2) As noted by the authors, the inter-sister CO increase was not expected and seems to contradict the previous publications by Adamo et al. and Bickel et al. Have the authors thought about testing intersister recombination with another allele of their mutants to exclude effects due to unknown mutations? This issue needs to be addressed and discussed.

We now include data with the ICR assay performed in both the classical *brc- 1(tm1145);brd-1(dw1)* mutant and null *smc-6(ok3294)* mutant. The ICR results in *brc- 1(tm1145);brd-1(dw1)* mutants revealed a clear increase in homolog-independent crossovers, corroborating our *brc-1(xoe4)* results (Figure 1 —figure supplement 2B). We observed both crossover and noncrossover repair outcomes in *smc-6(ok3294)* mutants at levels comparable to wild type, although *smc-6* mutants did not display a large increase in crossover events that we observed in *smc-5* mutants (Figure 2 —figure supplement 1B). Thus, our additional experiments affirm that neither SMC-5 nor SMC-6 is required for homolog-independent repair. While both *smc-5(ok2421)* and *smc-6(ok3294)* null mutants prevent SMC-5/6 complex assembly (Bickel *et al.*), it is possible that the residual proteins of the SMC-5/6 complex in either single mutant (*e.g.* SMC-5 protein in the *smc-6(ok3294)* null mutant) may still exert subtle functions sufficient to influence DNA repair outcomes that are independent of an assembled SMC-5/6 complex. We have therefore amended the text of our manuscript to specify that our results are specific to the function of SMC-5, rather than SMC-5/6.

Our multipronged assessment of DNA repair phenotypes provides further confidence that a strong modifier allele is not affecting our analysis of *smc-5* and *brc-1* mutants relative to previous work. In our hands, *smc-5(ok2421)* mutants closely recapitulated previously published RAD-51 foci counts (Figure 4B), sensitivity to ionizing radiation (Figure 4C), and defects in chromosome compaction described in Bickel *et al.* Similarly, *brc-1* mutants displayed ionizing radiation sensitivity (Figure 4C), disrupted RAD-51 loading in mid-late pachytene (Figure 4 —figure supplement 2C), and a high incidence of males (a product of chromosome nondisjunction) when maintained in stocks; phenotypes previously reported in Adamo *et al.*, Janisiw *et al.*, and Li *et al.*

Taken together, our data complements previous work on BRC-1 and SMC-5/6. Our experiments affirm the long-standing hypothesis that BRC-1 and SMC-5 regulate intersister DNA repair; however, our data indicate that they are not necessarily required for intersister repair (which is what was originally hypothesized), but instead our results support a growing body of evidence that BRC-1 and SMC-5/6 regulate diverse meiotic DNA repair outcomes. Our study therefore provides vital context to the meiosis and DNA repair fields. To emphasize and clarify these points (especially the fact our data complements prior published data), we have summarized these notions in a new section of our discussion (lines 545-589).

3) L303-304: The interpretation of the increased GFP progeny is a major issue and cannot be redirected to a few sentences in the methods section as this is the major topic of this study. The methods section outlines several alternatives, whereas the main text seems to prioritize some. This seems arbitrary and should be clarified with experimental data (for instance testing the SNP hypothesis).

We agree that the increased frequency of ICR assay recombination events in both *smc-5* and *brc-1* mutants is of potential biological importance, and we therefore performed experiments to test our hypotheses that the rate of increased recombinants in *brc-1* and *smc-5* mutants may be underpinned by relaxed heteroduplex rejection and/or increased Mos1 excision. It is important to note, however, that none of the conclusions made from the ICR assay in our manuscript are informed by the absolute frequency of ICR assay recombinants; we only draw conclusions from the ratios of homolog-independent noncrossovers and crossovers. We further make no claims as to the absolute frequency of intersister events that occur in *brc-1* and *smc-5* mutants.

We now include new experimental data where we tested whether altered heteroduplex recognition or rejection may influence the frequency of detectable ICR assay recombinants by specifically performing the ICR assay in a *msh-2* mutant, which is required for mismatch repair (Figure 1 —figure supplement 2, Chakraborty and Alani 2016). We found that the overall frequency of recombinants was not significantly altered in this context, disfavoring the mismatch-rejection model.

We next attempted to directly measure the frequency of Mos1 excision following heat shock by quantifying the number of RAD-51 foci produced following heat shock with the ICR assay in *spo-11*, *smc-5::AID::3xFLAG;spo-11*, and *brc-1;spo-11* mutants. However, we found that our *smc-5::AID::3xFLAG;spo-11* ICR assay strains were sick and mated very poorly, preventing us from obtaining sufficient animals to perform the experiment. Further, the *brc-1(xoe4);spo-11* ICR assay strains were genetically unstable and segregated phenotypes consistent with balancer breakdown. We hypothesize that this problem is due *brc-1* intolerance for large heterozygous genomic rearrangements, such as the nT1 translocation balancer required to properly balance *spo-11* and the ICR assay.

We attempted to balance *brc-1* with the multiply inverted balancer chromosome qC1 (Edgley *et al. Micropub.* 2021) and observed similar genetic instability. As BRC-1 represses heterologous recombination (Leon-Ortiz *et al. Mol. Cell* 2018), *brc-1* mutation may be broadly incompatible with this balancer class. Unfortunately, the smaller inversion balancer that covers the *unc-5* locus on chromosome IV (Dejima *et al. Cell Rep.* 2018) contains a transgene disrupting the *unc-5* locus, rendering it unusable for our purposes. We therefore were unable to test the absolute rate of Mos1 excision in the ICR assay in *smc-5* and *brc-1* mutants. As we noted in the text of our manuscript (Lines 191- 194,271-273), the maximum incidence of recombinants in the *smc-5* and *brc-1* IH assays were not elevated relative to wild type, suggesting that increased Mos1 excision is not increased in these mutants. Future work beyond the scope of this manuscript will be required to determine the underlying cause of the elevated rates of GFP+ ICR assay progeny in *smc-5* and *brc-1* mutants.

4) The authors cannot conclude that both assays (L 323 support a function for SMC-5 in repressing intersister crossover in *C. elegans* meiosis: Data from Edu is supported by only 6 bivalents coming from 2 oocytes. In addition, could the authors demonstrate that those exchanges are taking place during meiosis? (as performed for him-6, this may require to analyze a smc-5 spo-11 mutant).

We appreciate the reviewers’ comment and have quantified additional chromatid pairs from *smc-5* mutants. Our collaborators (David Almanzar and ofer Rog) identified additional SCE chromatids at a frequency in line with prior experiments (Figure 2D). We also appreciate that assessing for SCEs in an *smc-5;spo-11* mutant could be informative. However, due to the dissertation completion timeline of our collaborators, this analysis is beyond the scope of this manuscript. To interpret this data more conservatively, we have amended our text to emphasize that our *smc-5* SCE counts should be interpreted as evidence that meiotic and/or mitotic intersister crossovers do occur in the absence of SMC-5 but may not reflect the absolute levels of SCEs (Lines 262-266).

5) L361: the potential delay should be shown. Assigning the two time-windows as inter homolog and non interhomolog is at the basis of the experimental approach and whether it is altered in the mutants or not must be known and taken into account for interpretations.

We agree with the reviewers that the potential delay in meiotic progression is of interest, and recent work using the early prophase marker DSB-1 was recently published affirming that prophase I progression is delayed in *brc-1* mutants (Trivedi *et al. NAR* 2022).

However, our interhomolog assays (Figure 1 —figure supplement 3, Figure 2 —figure supplement 2) already serve as a key control that addresses this concern. In both *smc-5* and *brc-1* mutants, additional recombination events were not observed in the non- interhomolog window timepoint (10-22hr), indicating that the timepoints we use for the interhomolog/non-interhomolog windows are appropriate for the mutants assessed in this study.

6) Analysis of ICR and IH in brc-1 smc-5 double mutant. The authors could use balancers for both loci to maintain the strain, and minimize accumulation of genomic instability (from the strain list, a balancer was used for smc-5 but not for brc-1). Alternatively, the Smc-5 degron allele could be used.

We agree with the reviewers that the *brc-1 smc-5* double mutant is of interest. Unfortunately, the balancer options available for the *brc-1* locus of Chromosome III are limited to translocation balancers, and in our hands translocation balancers (both nT1 and qC1) are highly genetically unstable in a *brc-1* mutant background, limiting their application for these purposes. We therefore introgressed a germline-specific TIR1 transgene into the *smc-5::AID::3xFLAG* background, enabling germline-specific SMC-5 depletion in the presence of auxin (Figure 4 —figure supplement 2A,B). We then generated the *smc-5::AID::3xFLAG;brc-1(xoe4)* double mutant ICR assay strains and performed the assay following SMC-5 depletion (Figure 4A). We found that the ICR assay in *brc-1* mutants following SMC-5 depletion appear very similar to *brc-1* single mutants, suggesting that BRC-1 acts upstream of or in the same pathway with SMC-5 to regulate homolog-independent repair outcomes (Lines 344-364).

7) Irradiation experiments: Please provide cytological data of DSB repair detection (ie RPA or Rad-51) in the different mutant contexts and information about the number of DSBs induced at the different doses of IR (14 DSB per homolog pair for 5000 rads as from Almanzar et al?). Also provide information about the γ sensitivity of germ cells of wt, single and double mutants.

We appreciate the value of the reviewers’ suggestion to additionally assess DSB repair cytologically to complement our functional analysis. We examined RAD-51 foci in wild- type, *smc-5*, *brc-1*, and *smc-5;brc-1* germlines (Figure 4B). The results of this analysis are discussed in the manuscript at Lines 365-383. As we used γ radiation from a cesium source rather than X-rays to induce DSBs in this study, we reference the DSB induction estimates calculated in Yokoo *et al. Cell* 2012 that also used γ radiation to induce DSBs. From Yokoo *et al.*, a dose of 5000 Rads is expected to induce approximately 9.8 DSBs/chromatid pair, while a dose of 5000 Rads would inflict 19.6 DSBs/chromatid pair. We have included this information in the text for the readers’ reference (Lines 390-392).

The readout of the irradiation experiments is brood viability. One major problem with the interpretation is that sperm is also irradiated and can contribute to the phenotype. To avoid this issue, the authors should irradiate hermaphrodites and mate with non-irradiated males. This could be tested at least for one genotype (ie brc-1 smc-5) to evaluate the potential impact of sperm irradiation.

We appreciate this point made by the reviewers, particularly considering recent evidence that sperm-specific irradiation can cause intergenerational lethality in *C. elegans* (Wang *et al. Nature* 2023). Wang *et al.* demonstrate that even at doses of radiation much higher than those used in this study, the sperm-specific impact on brood viability is minimal in the F1 generation following P0 irradiation. The notable exception to this observation is in *polq-1* mutants, as DSBs in mature sperm DNA are repaired by maternally contributed POLQ-1 in the fertilized zygote. To assess the potential confounding effect of sperm to our brood viability experiments, we irradiated *smc-5;brc-1*, *polq-1*, and *smc-5;brc-1;polq- 1* mutant hermaphrodites and then mated these animals to unirradiated males (Figure 4 —figure supplement 3A,B, see Methods lines 781-783). In accordance with Wang *et al.*, introduction of *polq-1* mutation did increase the sensitivity of brood viability to radiation due to sperm-specific effects. However, mating did not rescue brood viability in the non- interhomolog window of *smc-5;brc-1;polq-1* mutants, supporting our conclusion that TMEJ is engaged in late stages of meiotic prophase I to resolve DSBs in *smc-5;brc-1* mutants. We have added description of these results to our Results section (Lines 447- 460).

8) The main figure (figure 4) should present brood viability, and the fit to the model as supplementary figure, as qualitatively the effects can be visualized by examining the brood viability: i) greater sensitivity of smc-5 compared to brc-1 (the claim for a greater sensitivity in the non-interhomolog window is not convincing as the differences of the γ parameter are quite small, may be excepted for the brc-1 mutant at 5000 rads; in addition, no statistical test is validating the differences),ii) dispensability of BRC-1 at early prophase in low IR, iii) suppression of smc-5 sensitivity by brc-1 in the interhomolog window, iv) sensitization by polq-1.

We appreciate the reviewers’ feedback on the presentation of our irradiation data and subsequent analysis. In response, we have exchanged the positions of our plots representing the experimental data we gathered and the outputs from our Bayesian modeling analysis (Figure 4C, Figure 4 —figure supplement 3A).

9) Discussion:The discussion is mostly reiterating the data from the results and should be more focused on the conclusions that can be drawn, novel points or questions to be addressed, and relating to current knowledge on Brc-1 and Smc-5 in other species.The question of chromosome fragmentation in Smc-5 and discussion about the data from this manuscript (L 674…) is unclear. How could the mutations detected in smc-5 (Volkova et al.) explain the efficient recovery of HR events by the ICR and IH assays? As indicated above (point 2), the authors should discuss the previous study concluding that Brc-1 is required for intersister recombination (Adamo et al., 2008).After citing this work in the introduction, the authors ignore this opposite conclusion from the discussion. May be the chromatid fragments in brc-1 msh-4 are products of TMEJ ? Similarly, Bickel et al. 2010, proposed that Smc-5 is required for homolog-independent (presumably intersister) events. The authors must solve this apparent contradiction.The question of potential differences between Spo11 and Mos1 induced breaks should be discussed and thus with alternative interpretations.

We appreciate the interesting points raised by the reviewers here. In reference to Adamo *et al.* and Bickel *et al.*, we have added additional context to our discussion (Lines 516- 527, see also our response to Major Revision point #2). We do not feel that our results contradict the Adamo and Bickel results and instead provide additional context to their data by applying technologies not available at the time of these original studies to directly test the specific role of SMC-5/6 and BRC-1 in homolog-independent DSB repair. Our parallel application of the EdU labeling assay (which monitors primarily SPO-11- induced DSB repair) and ICR assay (which monitors Mos1 excision-induced DSB repair) addresses the concern that our detection of intersister events may be unique to exogenous DSB repair. We have emphasized this point both in the Results (Lines 164- 165) and Discussion (Lines 524-527).

We thank the reviewers for their feedback on the structure of our Discussion. We regret that the reiteration of our conclusions, which we included with the intention to help guide a non-recombination expert reader through this section, deemphasized the primary discussion points we wished to convey. We have therefore streamlined the entire Discussion section to focus on contextualizing our results in the light of prior research.

10) The model presented in Figure 5, with the molecular intermediates is only indirectly supported by the data which does not allow determining which steps of the reaction are affected besides for the resection phenotype from the gene conversion tract analysis. The model would be better suited to be presented as a supplementary figure.

We have moved this figure to supplement (Figure 4 —figure supplement 6).

11) Box plots: Plot data points instead of box plots: Figure 1A, 1c, supfig2b, 2d, Figure 2B, 2d, Figure 3C, d

We appreciate the reviewers’ suggestions to improve the clarity of our data visualization. However, the panels indicated represent count data, and therefore the bars themselves accurately reflect the relevant proportions used for statistical analysis. As the ICR assay requires the quantification of low-frequency events (~0.7% of progeny in wildtype ICR assays) and the synchronization of P0 worms tested in the assay must be performed within a limited temporal window (see Methods lines 846-847), we sum together 4 replicates of the ICR assay or 2 replicates of the interhomolog assay when making comparisons between mutant contexts in either assay. The bar plot visualizations we employ and our approach for statistical analysis is consistent with previous work (Rosu *et al. Science* 2011, Almanzar *et al. Curr. Biol.* 2021, Toraason *et al. Curr. Biol.* 2021). For transparency in our results, we have updated the relevant source data files to specifically delineate the number and phenotypes of progeny assayed in each replicate.

Other revisions:12) Abstract: L21: SMC-5 was not shown to limit error prone repair.

This line has been removed.

13) The authors should introduce more widely the current knowledge about Brca1 and Smc5/6 with information from other species. Their results should also be integrated into a larger context.

We appreciate that our reviewers suggestion to contextualize our results broadly in the field. Our revised manuscript includes numerous references to work in yeasts and mammals that inform both our approach to this work and discussion of our conclusions. We have additionally added references to data from plants (Lines 539-541).

14) It is unclear whether each mutant was analyzed in parallel with a wild-type control. In principle to control for experimental variations (temperature…) this should be done. As presented in figures, one has the impression that wild-type and mutants were done in parallel. Please explain clearly the experimental procedures in the methods, and provide relevant information in the legends.

We thank the reviewers’ for identifying sections of our Methods that are unclear. To perform the time course of the ICR assay, we manually isolate all the parent hermaphrodites to be included in the experiment within a brief window precisely 16-18hr preceding heat shock. As the cross schemes required to generate some of P0 hermaphrodites yield the desired parent at a frequency as low as ~3 % (e.g. for *smc-5* mutants) and many parent hermaphrodites are required for these experiments, its is infeasible to perform these assays in parallel. We have added more transparent language to make this point in our Methods section (Lines 869-877) and in figure legends, where appropriate.

15) In several figures (ie Figure 1C and others), sample sizes and duplicates are not clearly indicated: how many animals, how many oocytes. It would be easier to have N values on the figures. It is mentioned in the methods that ICR assays were repeated 4 times with at least 20 parents each (what about the IH assays ?), but the source data shows only the total pool of progeny tested. What is the inter-parent and inter-replicate variation?

We appreciate the reviewers’ suggestions to improve the clarity of our methodology. We have added specific sample sizes to our figure legends where appropriate and provided details on the specific progeny counts in our supporting data for individual replicates.

We also thank the reviewers for catching our mistake in not including specific replicate information for the IH assays. The IH assay was performed twice for each genotype, as the high frequency of recombinants in the IH assay requires fewer progeny for robust quantification of recombination phenotypes relative to the ICR assay. To ensure that we correctly distinguished between noncrossover and mutagenic end joining events among these recombinants, we also had to individually sequence the *unc-5* locus of each potential noncrossover recombinant derived from our IH assays (461/483 non-Unc recombinants were successfully identified, and the remainder that couldn’t be sequenced were labeled as ‘undetermined’, Figure 1 —figure supplement 3, Figure 2 —figure supplement 2), reducing the throughput of these assays relative to previous work (Rosu *et al. Science* 2011, Toraason *et al. Curr. Bio.* 2021). This information is now included in our Methods (Lines 941-981). For Figure 2C, our collaborators (David Almanzar and Ofer Rog) provided additional numbers which are now included the text and figure legends.

16) For counting bivalents, some with visually interpretable bivalents were counted. Likely some bivalents either DAPI or Edu staining was difficult to interpret. How to exclude a selection bias in this analysis?

We acknowledge this concern from the reviewers. As described in the methods of Almanzar *et al. Curr. Biol.* 2020, the paper in which this technique was established: “… the poor resolution in the z axis prohibited analysis of chromosomes that were not oriented horizontally relative to the imaging plane. Exchanges (or lack thereof) were examined only in chromosomes where one of the two sisters was labeled with EdU and the two sisters were clearly distinguishable in the xy-plane. Sister exchanges were scored when EdU labeled chromatin crossed the chromosome axis (decorated with anti- HTP-3 antibodies), which represents the inter-sister interface. For bivalents, each of the long arms was scored independently.”

17) Fig1D: some numbers do not match the table: 1/41 instead of 1/43…, same for 1E.And the mutant non-unc is 17.2% in the figure but 13.2% in the text.

We thank the reviewers for catching our typo in the text. The revised manuscript has been amended to accurately reflect the 17.2% statistic. The 1/41 number in Figure 1D refers specifically to the number of successfully sequenced GFP+ progeny.

Unfortunately, we were unable to successfully sequence n=2 GFP+ animals identified in our *brc-1* ICR assays. We therefore excluded these animals from the calculation presented in Figure 1D.

18) Figure 3 A and C: The number of noncrossover conversion tracts in the non-IH window for smc-5 and brc-1 seems to be too low (with one long event in brc-1) to be statistically valid, even though a p value was calculated by the Fisher's exact test. The authors should dampen their conclusion that Brc-1 only affects tract length in the IH window or alternatively increase sample size.

We appreciate the reviewers’ point regarding this sample size. We have amended the text (Lines 306-308) to emphasize that our data only supports that the role of BRC-1 in regulating conversion tract length in the IH window but does not necessarily preclude it from performing this function in the non-IH window.

19)L 39: SPo11 references: Keeney et al. 1997; Bergerat et al. 1997.L 44: The use of "resolve" for DSB is odd (also L 55, 74).L 51: "In SDSA, one resected end" might be better than "the resected end".L 74: Utilize homologous recombination.

Fixed

L 104-5: vague; not additive could mean synergistic or unchanged.

We thank the reviewers for pointing out a place where our text was unclear. Both *smc-5* and *brc-1* mutants exhibit chromosome fragments (indicative of unrepaired DSBs at diakinesis). This phenotype is not additive nor synthetically increased in the double mutant, providing rationale that SMC-5 and BRC-1 may function in the same pathway. We have modified the text to better emphasize this point (Lines 98-100)

L 121 (and L514-515). For clarity, it would be good to use 'more genetic language. Eg 'the hypersensitivity of smc-5 mutations towards IR is suppressed/bypassed by brc-1 (mutation).

Fixed

L514: Could the term resilience be avoided, replaced by Eg:.…strains or more sensitive to IR than WT,.. are hypersensitive to IR,.. the IR sensitivity of smc-5 mutants is suppressed by brc-1….

Fixed

L 176-177: seem out of place, please revise.

We thank the reviewers for catching this typo. We have fixed the indicated text (Lines 135-153)

L179: "Does not necessarily represent recombinant progeny". The meaning is unclear. Please clarify.

Fixed

Line 198 and elsewhere: "is not required for" could be understood to mean "is not involved in." The intended meaning is "is not essential for."

Thank you for this clarifying suggestion, we have made the relevant changes to favor this language.

L 210. More frequently is vague please provide quantitative information.

We appreciate the reviewers’ suggestion to provide more quantitative language, however, the specific frequencies of mutagenesis in *brc-1* mutants seen in the very thorough Kamp *et al.* and Meier *et al.* studies vary depending on the mutagen and resulting mutations observed. To avoid including too much verbiage that could prove distracting to the reader, we have modified the text to better emphasize the important point that *brc-1* mutants incur mutations normally repressed during meiosis (Lines 187- 188).

L 246: "a slight delay in the rate of meiotic progression" the rate is not delayed.

Previous work (Rosu *et al. Science* 2011) indicates that mutants that impede crossover formation and therefore delay meiotic prophase progression extend the window in which the homolog is available as a repair template in the IH assay. Our results parallel recent cytological evidence using the early prophase marker DSB-1 (Trivedi *et al. NAR* 2022) that also indicates that *brc-1* mutants exhibit altered prophase progression. We have updated the text to reflect this relevant evidence (Results Lines 196-198, Discussion Lines 531-535).

L 261: "at the time of Mos1 excision" perhaps should say "at the time of heat shock" since there is likely some delay between heat shock and transposase expression and transposon excision.

We have amended our statement accordingly.

L 269 suppl figure 3B. It would be better to precisely refer to the events shown in the text. Eg label (1-4) and refer to (3B.1) 3B.2 etc etc.

We agree with this suggestion from the reviewers and have modified the figure and text accordingly.

L 283: 5 not 4 mutants have been identified (Figure 1E), but only 4 are reported?

We thank the reviewers for identifying this point of potential confusion. As described in our methods (Lines 966-969), we successfully identified two independent *unc-5* mutations in one of the non-Unc progeny derived from the IH assay. As somatic Mos1 excision can occur in the ICR assay (Toraason *et al. Curr. Biol.* 2021, Toraason *et al. STAR Protocols* 2021), these mutations likely stem from separate meiotic and somatic Mos1 excision events. We cannot distinguish which of these mutations are meiotically- derived and therefore wish to avoid making claims pertaining to the origin of this mutant. We have added a statement to the main text clarifying this point (Lines 220-221).

Line 299: "SMC-5/6 is therefore not required for viability in *C. elegans*…" this conclusion does not follow from the data presented in the preceding sentences.

*C. elegans smc-5, smc-6,* and *smc-5;smc-6* null mutants survive and are fertile despite ablation of the SMC-5/6 complex and can produce living progeny for many generations (Bickel *et al.* 2010), indicating that nematodes are capable of essential biological function (are viable) in the absence of this complex. In line with previous edits suggested by the reviewers, we have amended the text of this sentence to specify that SMC-5/6 is not essential for viability in *C. elegans*, rather than not required for viability (Line 237).

L 318: intersister in smc-5. "Among a limited sample", please clarify, does it refer to 6 individuals?

We have updated the text to specify the number of chromatids examined (Line 260).

Supplemental Figure 3B: using a monospaced font in the DNA sequences would ensure that the nucleotides in each strand were properly aligned.

We thank the reviewer for this suggestion. We have amended the figure accordingly.

Figure 2C-D: it would be helpful to see an image of an "uninterpretable" bivalent in the SCE assay and to know what percentage of bivalents were discarded as uninterpretable in wt, brc-1, and smc-5 animals.

The SCE assay results were performed by our collaborators (David Almanzar and Ofer Rog). They state, “most excluded chromosomes were to what are very technical as opposed to biological reasons (*e.g.* the stochasticity in EdU incorporation and progression of nuclei in the gonad; and the orientation of the bivalent to the xy-axis of the microscope).” Also, please see the response to Essential Revision #16 regarding the original publication of the SCE assay method developed by the Rog lab (Almanzar *et al., Current Biology* 2021). Due to the lack of EdU incorporation, uninterpretable bivalents were not imaged by the collaborators on the project.

Line 380: could be more specific. "in restricting/limiting intersister CO recombination".L 389: Figure 3A not 4A.

Fixed

L 428: This conclusion is confusing. Brca-1 mutant were shown above to have more CO (intersister). How can one conclude that it does not regulate CO recombination?

We thank the reviewers for identifying this opportunity for us to clarify our language. We intend to imply that our data does not support a function for regulating homolog- independent CO intermediates that would influence the extent of gene conversion following joint molecule resolution. We speculate in our discussion (Lines 522-577) as to how regulation of noncrossover-precursors may influence noncrossover/crossover DNA repair outcomes. The text has been edited to more clearly convey our intent (Lines 318- 320).

L 448: Where is the heteroduplex DNA information? Please provide this data.

Positions at which we detected heteroduplex DNA signals are depicted in the conversion tract plots as open circles, as outlined in the relevant figure legend. We acknowledge that these sites may be difficult to see, so we have added asterisks to indicate these tracts (Figure 3, Figure Legend 3).

Figure 3C and D (graphs): font is too small.

Fixed

Figure 4, supplemental figure 4: it would be interesting to see controls with for example polq-1 single mutants and polq-1 double mutants with brc-1 and smc-5. These controls might shed some light on what is going on and get closer to mechanism.

We agree with this suggestion from the authors and have assessed *polq-1* single mutant sensitivity to ionizing radiation (Figure 4 —figure supplement 3) in both unmated and mated conditions. In line with previously published work (Kamp *et al. Nat. Comm.* 2020), we demonstrate that POLQ-1 contributes minimally to meiotic DSB repair when BRC-1 and SMC-5/6 are present.

Sup Figure 4: What is the sample size ? Wild type: no point is visible; brc-1, only 1 point is visible at 0 rad ? Please clarify. In panel B, the simulation is probably the distribution shown in black, please clarify.

We thank the reviewers for indicating how we could improve our data visualization. The low variance in unirradiated wildtype animals caused all the datapoints to cluster together in Figure 4C. To better visualize the brood viabilities of the individual hermaphrodites, we changed the shape of the datapoints and added horizontal jitter (Figure 4C, Figure 4 —figure supplement 3B). Additionally, we have added clarifying text to our posterior simulations to emphasize the experimental (formerly labeled ‘empirical’) and simulated distributions (Figure 4 —figure supplement 3C).

In some cases (smc-5 2500 rad) the simulation does not fit the data. This is problematic because the validity of the γ parameter is therefore uncertain. The source data do not show the individual data points (ie brood viability of individual hermaphrodites), please revise.

We appreciate the reviewers’ pointing out instances where our model fit could be improved. By reparametrizing our model to use the mean and shape parameter sum of the Β distribution, we improved the model fit (Methods Lines 794-802, Figure 4 —figure supplement 3C).

Supplemental Figure 7 is never referred to in the manuscript.

We included this figure (now Figure 4 —figure supplement 2 in the revised manuscript) to demonstrate that RAD-51 localization occurs using our tagged *smc-5* allele, in contrast to the known RAD-51 loading defect in *brc-1* mutants. Supplemental Figure 7 was referenced on Line 596 of the original manuscript and is referenced on line 354 in the revised manuscript.

What is smc-5 (phx3065) ? PHX3065 seems to be a strain name.

We cannot find the place in the manuscript where we refer to *smc-5(phx3065)*. As listed in our Methods (Line 754 in the revised manuscript), the strain PHX3065 carries the *smc- 5(syb3065)* allele, which is the endogenously tagged *smc-5::AID*::3xFLAG*. If the reviewers would please identify the position where we misannotated this genotype, then we would be happy correct this mistake.

L 642: "influences noncrossover intermediate processing" is vague.

We have altered this line to be more specific as a part of the larger requested changes to our discussion (Lines 507-509).

Are the most important strains for measuring inter-sister and interhomolog recombination deposited in the GCG strain collection?

The ICR assay strain (DLW14) and repair template strain (CB791) are available through the CGC. The strain IH assay strain (AV554) was developed in the Villeneuve Lab as part of Rosu *et al., Science* 2011 and we were unaware that it was not added to the CGC strain collection with the other strains from that published paper (many thanks to the reviewers for pointing this out). We have recently submitted AV554 to the CGC so that it can be easily obtained by the *C. elegans* community.

Reviewer #1 (Recommendations for the authors):Figure 1: The number of biological duplicates is unclear and single-point data cannot be found in the tables. It is said that assays were replicated 4 times and at least 20 parents each, but sup table 1 only show the total progeny. What is the inter-parent and inter-replicate variation?

The individual replicate numbers of these experiments are now available in the supporting data associated with the relevant figures.

Fig1D: some numbers do not match the table: 1/41 instead of 1/43.…, same for 1EAnd the mutant non-unc is 17.2% in the figure but 13.2% in the text ?

We thank the reviewer for catching this discrepancy. We have thoroughly vetted the manuscript to ensure all numbers are correct. Also, please see our response to Essential Revision #17.

L171: "Does not necessarily represent recombinant progeny". The meaning is unclear. Please clarify.

Fixed

L 222. More frequently is vague please provide quantitative information.

Fixed

L 252: 5 not 4 mutants have been identified, but only 4 are reported?

See our response to Essential Revisions #19.

Interhomolog assay: the progeny is referred to as non-unc or unc-5 (+). This is confusing , please use one of the two. It seems that referring to unc-5 (+) is better even if this can include some mutational events.

We appreciate the reviewers’ suggestions on our specific language when referring to phenotypically revertant progeny in the IH assay, and had previously considered using the suggested *‘unc-5(+)’* nomenclature when drafting our manuscript. The term ‘non-Unc’ is appropriate terminology commonly used by *C. elegans* geneticists to refer to nematodes with normal locomotion. Our rationale to use this language, rather than *unc- 5(+),* is to avoid confusion over the distinction between recombinant and mutant progeny, as this is an important delineation in repair events that informs our conclusions. Thus, while we greatly appreciate this feedback from the reviewer, we have maintained our original language in this revised draft.

L228 and sup2A: increase of interhomolog !!

Please see our response to Essential Revision #2 and the associated changes to the Discussion.

L287: The interpretation of the increased GFP progeny is a major issue and cannot be redirected to a few sentences in the methods section as this is the major topic of this study. The methods section outlines several alternatives, whereas the main text seems to prioritize some. This seems arbitrary and should be clarified with experimental data (for instance testing the SNP hypothesis).

Please see our response to Essential Revision #3.

L324: intersister in smc-5. "Among a limited sample", please clarify, does it refer to 6 individuals?

We have added relevant sample size information to the text and figure legends.

L 232: What are the other defects?

We are uncertain what this comment is in reference to but would be happy to rectify any concerns the reviewer has that this line is in reference to.

L338: the potential delay should be shown.

Please see our response to Essential Revision 5.

L343: It is not because the mutant has no defect in the IH homolog assay that smc-5 does not have a function in IH repair. Please revise.

We have amended the statement accordingly.

Figure 2: A and B: why 2 graphs: the data 10-22hr is plotted twice.

Our ICR/IH data can be examined both by individual timepoints and overall within the IH/non-IH window. As there is only one timepoint in the non-IH window, there is no distinction between these two representations as the reviewer notes. However, this alternative representation can provide information in particular for the IH window, and for clarity of information we provide both visualizations for the reader. We have previously used this style of visualization in Toraason *et al. Curr Biol. 2021*.

L365: Figure 3A not 4A

Fixed

Figure 3C and D (graphs) : letter sizes are too small

Fixed

L388: The numbers for non-crossovers in the IH window is too low (8 events) to draw a convincing comparison between wt and brc-1.

Please see our response to Essential Revisions #4.

L402: This conclusion is confusing. Brca-1 mutant were shown above to have more CO (intersister). How can one conclude that it does not regulate CO recombination?

Please see our response to Essential Revisions #19.

L422: Where is the heteroduplex DNA information? Please provide.

We have amended Figure 3 and its associated figure legend.

Sup Figure 4: What is the sample size ? Wild type: no point is visible; brc-1, only 1 point is visible at 0 rad ? Please clarify. In panel B, the simulation is probably the distribution shown in black, please clarify. In some cases (smc-5 2500 rad) the simulation does not fit the data. This is problematic because the validity of the γ parameter is therefore uncertain. The sup. table 3 and 4 do not show the individual data points, please revise.

Please see our response to Essential Revisions #19.

What is smc-5 (phx3065) ? PHX3065 seems to be a strain name.

Please see our response to Essential Revisions #19.

Figure 4: A DSB repair defective control would be important to evaluate how severe the defect in smc-5 mutant is, may be Rad-51 or Rad-54 mutants could be used. The conclusion that the sensitivity (estimated by Γ parameter, 4A) is greater in the non-interhomolog window is not supported by a statistical test. Please revise.

We appreciate this suggestion from the reviewer. Both RAD-51 and RAD-54 null mutants are sterile due to severe chromosome fragmentation, as all meiotic recombination in *C. elegans* is dependent upon these proteins. We also appreciate the feedback from the reviewer on our statistics. To account for multiple levels of variation (including inter- hermaphrodite and inter-genotype variation in brood size and baseline brood viability) into our analysis, we employed a Bayesian statistical modeling approach and therefore compared the Credible Intervals of our parameter estimates to inform our conclusions.

The authors should analyze the phenotypes of the brc-1 smc-5 double mutant with the ICR and IH assays.

Please see our response to Essential Revisions #6.

Also a more direct monitoring of DSB repair must be done, for instance by detection of RPA and/or RAD51 foci in the different mutants (especially double and triple mutants) with and without the different doses of radiation. How many breaks are induced by 2500 and 5000 rads ?

Please see our response to Essential Revisions #7.

L 40: SPo11 references: Keeney et al. 1997; Bergerat et al. 1997

Fixed

L 45: The use of "resolve" for DSB is odd (also L 58)

Fixed

L75: utilize homologous recombination

Fixed

Comment on the DiscussionThe discussion is mostly reiterating the data from the results and should be more focused on the conclusions that can be drawn, novel points or questions to be addressed.The question of chromosome fragmentation in Smc-5 and discussion about the data from this manuscript (L 674…) is unclear. How could the mutations detected in smc-5 (Volkova et al.) explain the efficient recovery of HR events by the ICR and IH assays?The authors should discuss the previous study concluding that Brc-1 is required for intersister recombination (Adamo et al., 2008). After citing this work in the introduction, the authors ignore this opposite conclusion from the discussion. May be the chromatid fragments in brc-1 msh-4 are products of TMEJ ?Similarly, Bickel et al. 2010, proposed that Smc-5 is required for homolog-independent (presumably intersister) events. The authors must solve this apparent contradiction. Differences between assays and/or Spo11 vs Mos1 induced breaks or other features should be tested.The model presented in Figure 5 is not supported by the data which does not allow determining which steps of the reaction are affected beside a support for a resection phenotype from the gene conversion tract analysis. The authors could speculate on the different alternatives for the activities of Brc-1 and Smc-5 (and they are many) but presenting a synthetic model is misleading not coherent with the content of the data. It should be removed. If the authors want to keep this visualization, it should be in supplementary data.A schematic representation of the key assays: the meiotic progression in the gonad to show the reverse kinetic principle, the ICR and the IH assays as a sup figure would help understanding the data.

We thank the reviewer for this valuable feedback. We have addressed all of these comments in our revised manuscript as well as our responses to the Essential Revisions. In particular, please see our response to Essential Revisions #1, 2 which address most of the major points brought up in the above comments.

Reviewer #2 (Recommendations for the authors):Importantly, data on sister chromatid repair by CO and non-CO data are both supported by GFP reporter assays as well as by directly observing sister chromatid exchanges of EDU-marked chromatids. Having this said, the reviewer struggled to deduce how the reporter system used, allows for distinguishing CO and non-CO outcomes. Could a Suppl model; Figure showing the detailed intermediates be provided? I did not find this in the original *C. elegans* paper either.

We thank the reviewer for this suggestion and have added Figure 1 —figure supplement 1 in the revised manuscript to meet this need.

Would it be possible to analyze sister chromatid recombination (proportion of CO versus non-CO outcome as measured by reporter assay) also in brc-1 smc-5 double mutants?

Please see our response to Essential Revision #6.

A weakness in the study is that key phenotypes are only analyzed by one allele of brc1 and smc-5 each. Would it be possible to repeat a key experiment, eg EDU labelling or GFP reporter with a second allele? This way, the possibility of secondary, possibly unlinked mutations contributing to phenotypes can be formally excluded. Nowadays, a simple genome edit (generating a second allele) might do the trick.

Please see our response to Essential Revisions #2.

In the radiation sensitivity assays, would it be possible to also include an analysis of sensitivities of single and double mutants of mitotically proliferating germ cells (before the enter prophase). Such analysis would very much add to our understanding of differential pathway usage of germ cells.

Please see our response to Essential Revisions #7.

Figures 1-3, I think it would be good, wherever possible, in graphs and diagrams (Figure 1a, c, Figure 2a, b, Figure 3 c, d) to show individual data points in addition to error bars.

Please see our response to Essential Revisions #11.

Line (46) Is the statement that the majority of meiotic DSBs are repaired by interhomolog recombination true, also for *C. elegans*?

While the low frequency of SCE events as seen by Almanzar *et al.* suggest that intersister repair is not readily engaged in *C. elegans* meiosis, we acknowledge that the direct proportion of DSBs resolved by IH repair is not known in *C. elegans*. We have added clarifying text to this sentence (Line 48, 70-71).

Lines 121. I think, for clarity, it would be good to use 'more genetic language. Eg 'the hypersensitivity of smc-5 mutations towards IR is suppressed/bypassed by brc-1 (mutation).

Fixed.

Generally, whenever possible only describe mutant phenotypes, or protein function but not both within a sentence.

We appreciate this suggestion and have implemented it in the revised manuscript.

Line 269 suppl figure 3B. It would be good to precisely refer to the events shown in the text. Eg label (1-5) and refer to (3B.1) 3B.2 etc.

Fixed.

Line 403, Figure 3C and D. Be uniform in what you label as short and long conversion tracks.

We are examining separate classes of recombination events in these panels that display differing properties in a wild type context. Our delineation of short and long tracts reflects these baseline differences in order to facilitate comparisons to mutants that might alter mechanisms of NCO and/or CO gene conversion specifically.

Lines 514, and 515, same as the above comment, line 121.

Fixed.

Page 25 consider rewording. Can the word 'resilience' be avoided?

Fixed.

Eg:.…strains or more sensitive to IR than WT,.. are hypersensitive to IR,.. the IR sensitivity of smc-5 mutants is suppressed by brc-1…Line 587. 599 can this section be reworded to avoid the word 'normal'? Eg BRC-1 localization did not change upon IR in…..

Fixed.

Are the most important strains for measuring inter-sister and interhomolog recombination deposited in the GCG strain collection?

Please see our response to Essential Revisions #19.

Reviewer #3 (Recommendations for the authors):As currently written, the manuscript is very focused on *C. elegans* meiosis. The experiments are generally well done, and the results will be of interest to investigators studying DNA repair and meiotic recombination in *C. elegans*. However, it is unclear whether the results are unique to *C. elegans*, and the authors do not give sufficient context to convincingly frame their conclusions as novel or identifying conserved meiotic mechanisms.At least in the merged PDF of the manuscript, lines 176 and 177 seem out of place, and reading from line 147 (before Figure 1) to line 176 (after Figure 1) does not make sense. I think lines 176-77 should be deleted, but do they belong somewhere else? Also, the first reference to Figure 1A comes later in the text than I would have expected, so I'm not sure if other text was missing that should have been there.

We thank the reviewer for catching this formatting error. The deleted text has been restored in the revised draft.

The ICR and IH assays are so important to this paper that there should be a supplemental figure showing how they work and give different outcomes for CO vs NCO, and where the polymorphisms are. Even though they have been published elsewhere, it would be helpful not to have to find those papers to be able to follow the experiments done here.

We thank the reviewer for this feedback. The relevant recombination assays are now included as Figure 1 —figure supplement 1.

Line 461: How do the doses of IR used affect the number of DSBs or RAD-51 foci? Are these doses expected to increase breaks 2-fold? 10-fold? The experiments are difficult to interpret without knowing how DSB number (or some DSB proxy) changes.

Please see our response to Essential Revisions #7.

As the authors acknowledge, there is substantial variation in viability between the broods of different irradiated (and even unirradiated) hermaphrodites with the same genotype. The cause of this variability is not known. However, it is possible that sperm defects contribute to the variability. Irradiation of young adult hermaphrodites will induce mutations both in the germline nuclei undergoing meiosis and in the several hundred mature sperm made earlier in development. Thus, the "reverse timecourse" following irradiation is capturing both the effects of mutations made at different stages of nuclei undergoing oogenic meiosis and sperm that had already completed meiosis when irradiated. Larval lethality could result from either, and may be revealing how an embryo copes with inheriting damaged DNA from the sperm. It is also formally possible that sperm differences account for the variability observed in unirradiated worms. It is already difficult to compare the data using Mos1 induced breaks to the irradiation data because the assays are completely different. However, the Mos data also presumably is only measuring recombination during oocyte production (i.e. not in sperm), while larval lethality following irradiation is likely measuring damage in sperm as well as during oogenesis. I would recommend mating irradiated hermaphrodites with unirradiated males to simplify the analysis by looking only at consequences of damage that will ultimately affect the oocyte.

Please see our response to Essential Revisions #7.

The irradiation/larval lethality assay was used instead of the ICR/IH assays because "genetically balanced smc-5;brc-1 double mutants can still acquire mutations and become progressively sterile over the course of a few generations" (line 456). From the strain list, it looks like only smc-5 was balanced, but brc-1 was maintained in a homozygous condition. Given the availability of numerous small inversion balancers made by the Mitani lab, it should be possible to balance both loci. This may prevent the accumulation of defects and allow the ICR/IH assays to be done in double mutants. Alternatively, perhaps the degron-tagged smc-5 allele could be used to deplete SMC-5 acutely. If so, this would greatly strengthen the paper.

Please see our response to Essential Revisions #6.

For me, the way the irradiation data are presented calls into question their robustness, or at least suggests that the conclusions are only as good as the mathematical model. I think many of the key results are visible in Supplemental Figure 4A, at least at some doses. As a biology person and not a statistics person, I would find it substantially more convincing to discuss the empirical data first, then bring in the model data to support the empirical data. At the very least, I need better justification for the design of the statistical model and its reliability.

We appreciate this valuable feedback from the reviewer and wish to emphasize that it was not our intent to obscure our results. We agree with and appreciate the feedback that many of the phenotypes we noted are quite clear even from the raw data without any analysis, and have presented the experimental data in Figure 4C and moved our statistical analysis to Figure 4 —figure supplement 3.

The Βeta-Binomial model is commonly employed in statistical analysis of discrete observations, as it accounts for multiple levels of variation that may exist within a given dataset. Binomial distributions describe the incidence of discrete outcomes (*e.g.* whether the egg laid by a nematode will hatch or not) that occur with a probability of p. In a Βeta-Binomial model, p is not a fixed value and is instead described by a Βeta distribution.

This framework is useful for our analysis, as some DNA repair mutants exhibit high inter-hermaphrodite variation in baseline brood viability (in particular loss-of-function mutants deficient for multiple repair pathways, such as *brc-1;polq-1* mutants, as observed both in our study and by Trivedi *et al. NAR* 2022). In this context, the variance between hermaphrodites can be accounted for by the Βeta distribution in the model, while the proportion of dead progeny that are observed in a given brood is accounted for by the Binomial level of the model. We have taken a Bayesian approach to fitting this model to our data and have estimated values for the parameters of this model that reasonably recapitulate the data we observe and facilitate our inference in the effect of irradiation on brood viability.

Identical data are presented as wild-type controls in several figures (Figures1A and 4A, 1B and 4C, Supplemental Figure 1A and B, and elsewhere). This gives the false impression that each mutant was analyzed in parallel with a wild-type control. However, it is unclear from the text, figure legends, and methods whether any of the experiments were done with parallel controls. In fact, the wild-type dataset was performed once and combined with previous data (line 950). Ideally, wild-type and mutants would be analyzed at the same time, on the same batch of plates, same batch of worm food, and in the same incubator, since meiosis is temperature sensitive, and heat shock genes can be induced by a variety of stressors, including wet plates. If this was not done, the authors should clearly state it in the methods and ideally also in the figure legends, since it is an important point. I also think it would be less confusing, and would potentially make the paper easier to read, if some figures were combined, for example 1A and 4A could be one figure (with a legend stating that multiple replicates were done, but not always in parallel for different genotypes); 1B and 4C could easily be combined.

We thank the reviewer for this feedback. Please see our response to Essential Revisions #14.

Figures 1B and 4C: the N values are given for number of bivalents analyzed. The number for WT seems reasonable, but the numbers for mutants seem small even though this is a somewhat difficult experiment. How many bivalents were analyzed in each oocyte, and how were they chosen? How many oocytes were analyzed? How many animals?

Please see our response to Essential Revision #17.

Figure 3A and C: it is hard to interpret the data showing a statistically significant effect of brc-1 depletion on resection tract length because so many fewer NCO tracts were analyzed in brc-1 than in wt or smc-5 and only one long tract was identified. I guess this would only mean that the magnitude of the effect is being underestimated, not the opposite.

Please see our response to Essential Revisions #18.